# ANARCHIC FEDERATED BILEVEL OPTIMIZATION

## ABSTRACT

Rapid federated bilevel optimization (FBO) developments have attracted much attention in various emerging machine learning and communication applications. Existing work on FBO often assumes that clients participate in the learning process with some particular pattern (such as balanced participation), and/or in a synchronous manner, and/or with homogeneous local iteration numbers, which might be hard to hold in practice. This paper proposes a novel Asynchronous Federated Bilevel Optimization (AFBO) algorithm, which allows clients to 1) participate in any inner or outer rounds; 2) participate asynchronously; and 3) participate with any number of local iterations. The proposed AFBO algorithm enables clients to flexibly participate in FBO training. We provide a theoretic analysis of the learning loss of AFBO and the result shows that the AFBO algorithm can achieve a convergence rate of $\mathcal{O}(\sqrt{\frac{1}{T}})$, which matches that of the existing benchmarks. Numerical studies are conducted to verify the efficiency of the proposed algorithm.

## 1 INTRODUCTION

Federated learning (FL) is an emerging privacy-preserving training paradigm over distributed networks (McMahan et al., 2017). In FL, multiple edge devices (or clients) computing together to train a global model under the coordination of a central server. Instead of sharing local data with the central server, clients compute locally and only transmit the updates (or parameters) to the server. This paradigm is increasingly attractive due to the growing computational power of edge devices and the increasing demand for privacy protection. However, federated learning is facing more challenges than traditional distributed optimization due to the high communication cost, data heterogeneity (not identically and independently distributed, Non-IID), heterogeneity in clients' computing and communication ability, and privacy concerns. In recent years, (Karimireddy et al., 2020; Wang & Joshi, 2021; Yang et al., 2022a) have addressed most of the aforementioned challenges in the algorithmic design.

Bilevel Optimization (BO) consists of two classes of optimization tasks (Colson et al., 2007), one optimization problem nested within the other. The outer optimization problem is often called the leader's (upper-level) optimization problem, and the inner optimization problem is often called the follower's (lower-level) optimization problem. This two-stage optimization can be viewed as a constrained optimization problem, where the lower-level optimization problem can be viewed as the constraints of the upper-level optimization problem. It has been widely used in many important applications, such as meta-learning (Rajeswaran et al., 2019), hyper-parameter optimization (Franceschi et al., 2018), model selection (Kunapuli et al., 2008), adversarial networks (Creswell et al., 2018), game theory (Stackelberg & Peacock, 1952), and reinforcement learning (Sutton & Barto, 2018). Recently, some studies have provided asymptotic analyses to bilevel optimization, such as BSA (Ghadimi & Wang, 2018), TTSA (Hong et al., 2020), ALSET (Chen et al., 2021) and so on.

Federated Bilevel Optimization (FBO) is to run BO problems in the FL system. In this way, it allows a large number of clients to perform computations in parallel, rather than in series or with one client, which can not only save training time but also keep the data privacy. (Ghadimi & Wang, 2018; Hong et al., 2020; Liao et al., 2018) proposed methods to compute the estimated hyper-gradient. After that, FBO has received much attention (Huang et al., 2023) (Tarzanagh et al., 2022) (Li et al., 2022). For example, (Fallah et al., 2020) studied the federated meta-learning problems, (Khodak et al., 2021)

provided a federated hyperparameter optimization approach, and (Zeng et al., 2021) improved the fairness of federated learning using a bilevel method. In addition, (Chen et al., 2023) studied the decentralized stochastic bilevel problem, and (Jiao et al., 2023) proposed an asynchronous distributed bilevel algorithm.

To fully realize the potential of FBO, several challenges need to be addressed, such as heterogeneous and/or time-varying computation and communication capabilities of clients, and the clients' partial and/or asynchronous participation. First of all, clients' heterogeneous computation and communication capabilities lead to a large waiting time to use synchronous algorithms. Furthermore, if the fast clients (large computation rate or small communication time) run more local iterations, then the overall objective will converge to a local optimal instead of a global optimal. In addition, clients may not be able to participate in every round of the entire learning process.

**Our contributions**. In this paper, we provide a novel Anarchic Federated Bilevel Optimization (AFBO) algorithm to address flexible asynchronous participation, heterogeneous local iteration numbers, and non-IID data in federated bilevel problems. In particular, AFBO imposes minimum control on how clients participate in FBO by allowing them to 1) participate in any outer or inner rounds (respectively); 2) participate asynchronously across outer rounds or inner rounds (respectively); 3) participate with any numbers of local iterations; 4) participate with arbitrary dataset distributions. By giving maximum freedom to clients, AFBO enables clients to participate flexibly according to their heterogeneous and time-varying computation and communication capabilities. In the meanwhile, the asynchronous communication structure of AFBO can greatly reduce the wall-clock time, as clients synchronize their local models with the server by the most recent data stored in the server instead of waiting for synchronizing from stragglers. Our contributions are summarized as follows.

- We propose AFBO, a double-loop scheme Anarchic Federated Bilevel Optimization (AFBO) algorithm, which allows clients to participate asynchronously in any rounds with heterogeneous local iteration numbers and dataset distributions. The proposed AFBO algorithm allows clients to participate in FBO efficiently and flexibly with their heterogeneous system parameters and datasets. The algorithm design of AFBO involves some key techniques, including using the most recent local gradients of clients, and adjusting learning rates when the global model no longer fits clients' local data.

- We provide a convergence analysis for the AFBO algorithm. AFBO can be applied to bilevel optimization with non-convex upper-level and convex lower-level objectives. Our results show that the AFBO algorithm can achieve a convergence rate of $\mathcal{O}(\sqrt{\frac{1}{T}})$ as well as a linear convergence speedup ($\mathcal{O}(\sqrt{\frac{1}{mT}})$), which matches that of existing benchmarks. The results also characterize the impacts of clients' local iteration numbers, local model delays, and global model delays on learning loss.

- We conduct numerical experiments to verify the effectiveness of the proposed AFBO algorithm. The experimental results demonstrate the efficiency of the proposed algorithms.

The remainder of this paper is organized as follows. Section 2 reviews related work. In Section 3, we present the system model and algorithm design of AFBO. In Section 4, we provide the convergence analysis of the AFBO algorithm. Numerical results and conclusions are provided in Section 5 and Section 6, respectively.

## 2 RELATED WORK

### 2.1 FEDERATED LEARNING

FL has emerged as a disruptive computing paradigm for machine learning by democratizing the learning process to potentially many individual users using their end devices (McMahan et al., 2017; Bonawitz et al., 2019; Stich, 2019; Li et al., 2020; Wang & Ji, 2022). The past few years have witnessed tremendous research on FL. (Li et al., 2020; Zhu et al., 2021; Yang et al., 2022b; Wang & Ji, 2022) studied FL where only some of all clients participated in learning in a round. Most of these studies assumed that clients' participation is balanced (e.g., the set of participating

clients is randomly selected from all clients), such that each client has the same probability of participation (Li et al., 2020; Zhu et al., 2021). It has been shown that FL algorithms can achieve a vanishing convergence error with such an assumption. (Li et al., 2020; Zhu et al., 2021; Wang & Ji, 2022) studied synchronous algorithms where participating clients perform local computations and exchange local models in the same round (i.e., synchronous FL can also have partial client participation). (Mitra et al., 2021) proposed FedLin to address objective and systems heterogeneity in FL. However, synchronous algorithms can be inefficient as some clients may have to wait for other clients to complete their computations and/or communications, especially when there are stragglers due to heterogeneous clients' systems. Asynchronous algorithms are more efficient since a client can start its local computations in one round while completing the communication of its local model in another round (Lian et al., 2015; Yang et al., 2022b). (Yang et al., 2022b) has proposed the AFL algorithm, where clients can participate in a round or not in an asynchronous manner with different local iteration numbers. (Yang et al., 2022b) considers a single-level problem and AFBO works on a bilevel problem. From algorithm design aspects, AFBO is a distributed FBO algorithm, it needs to estimate the federated hyper-gradient where AFL does not need to. From the analysis perspective, AFL just has a single loop where AFBO needs to analyze both the inner loop and outer loop, which makes the analysis non-trivial. As not all problems can be formulated as a one-layer problem, e.g., meta-learning, etc., this paper considers a BO problem in the asynchronous federated distributed system setting.

## 2.2 BILEVEL OPTIMIZATION

The bilevel optimization problem is first introduced by (Bracken & McGill, 1973). Some recent works (Zhang et al., 2022) assumed there is an analytical solution to the lower-level optimization problem, and then the bilevel optimization problem can be reduced to a single-level problem. However, it is not always possible to find an analytical solution for lower-level problems. (Sinha et al., 2017) replaced the lower-level optimization problem with optimal surrogate under some sufficient conditions (e.g., KKT conditions). Then, the bilevel problem can be reformulated as a single-level constrained optimization problem. However, the resulting problem could be hard to solve since it often involves a large number of constraints (Ji et al., 2021; Gould et al., 2016). Then, (Ghadimi & Wang, 2018; Hong et al., 2020; Liao et al., 2018) proposed gradient-based methods, which compute the hyper-gradient (or the estimation of hyper-gradient), i.e., $\frac{\partial F(x,y)}{\partial x} + \frac{\partial F(x,y)}{\partial y}\frac{\partial y}{\partial x}$, and use gradient descent (GD) or stochastic gradient descent (SGD) methods to solve the bilevel optimization problems. As far as we know, most BO papers focus on synchronous communication with homogeneous local iterations in inner rounds, which is limited in real-world applications.

## 2.3 FEDERATED BILEVEL OPTIMIZATION

Most of the existing bilevel optimization algorithms focus on centralized settings and require collecting a massive amount of data from distributed edge clients. This may give rise to data privacy risks and communication bottlenecks (Subramanya & Riggio, 2021). In federated bilevel problems, it is challenging to approximate the hyper-gradient (i.e., $\frac{1}{M}\sum_{i=1}^{M}\frac{\partial F_i(x,y)}{\partial y}\frac{\partial y}{\partial x} \neq \frac{\partial F(x,y)}{\partial y}\frac{\partial y}{\partial x}$). (Okuno et al., 2018; Yu & Zhu, 2020) proposed automatic machine learning, which is a powerful tool for approximating the hyper-gradient. (Huang et al., 2023) studied data heterogeneity in federated bilevel problems under a synchronous setting with one local iteration in the inner loop. (Yang et al., 2023) provided an algorithm ShroFBO to reduce communication costs and allow heterogeneous local computation in the synchronous FBO system. (Huang, 2022) used momentum-based variance reduced local-SGD to reach a communication-efficient FBO. (Huang et al., 2022) proposed a different FL algorithm based on Local-SVRG to obtain exact gradient information and achieve lower communication complexity. (Zhang et al., 2021) proposed a new backward updating mechanism to collaboratively learn the model without privacy leakage in an asynchronous vertical federated learning system. (Li et al., 2023) provided some low communication complexity algorithms to solve FBO problems through the variance-reduction technique. (Jiao et al., 2023) proposed the ADBO algorithm to solve the bilevel optimization problem in an asynchronous distributed manner. This paper considers a federated bilevel optimization problem with *partial participation*, and *heterogeneous local iteration numbers* in an *asynchronous* federated communication setting-AFBO, which has several major differences compared to general BO works. In addition, our paper considers a

two-level optimization problem (e.g., meta-learning), which is different from the existing FL works (e.g., AFL).

## 3 ANARCHIC FEDERATED BILEVEL OPTIMIZATION

### 3.1 SYSTEM SETTING AND PROBLEM FORMULATION

Let's consider a general bilevel optimization in a federated learning system, which takes the following formulation.

$$\min_{x \in \mathbb{R}^p} \Phi(x) := f(x, y^*(x)) := \frac{1}{m} \sum_{i \in \mathcal{M}} f_i(x, y^*(x))$$

$$s.t. \ y^*(x) = \text{argmin}_{y \in \mathbb{R}^q} g(x, y) := \frac{1}{m} \sum_{i \in \mathcal{M}} g_i(x, y) \tag{1}$$

where $f_i(x, y) = \mathbb{E}\left[f_i(x, y; \xi_i)\right]$, $g_i(x, y) = \mathbb{E}\left[g_i(x, y; \zeta_i)\right]$ are stochastic upper- and lower-level loss functions of a client $i$, $\mathcal{M}$ is the set of clients, and $y^*(x)$ is the optimal solution of the lower-level problem. The goal of problem 1 is to minimize the objective function $\Phi(x)$ with respect to (w.r.t.) $x$, where $y^*(x)$ is obtained by solving the lower-level minimization problem.

The key step to find the solution of problem 1 is to exactly estimate the $\nabla \Phi$, where

$$\nabla \Phi(x) = \nabla_x f(x, y^*(x)) - \nabla_{xy}^2 g(x, y^*(x)) \left[\nabla_{yy}^2 g(x, y^*(x))\right]^{-1} \nabla_y f(x, y^*(x)) \tag{2}$$

It's hard to find the $y^*(x)$ in each step, and thus we usually use a surrogate

$$\bar{\nabla} \Phi(x^t) = \nabla_x f(x^t, y^{t+1}) - \nabla_{xy}^2 g(x^t, y^{t+1}) \left[\nabla_{yy}^2 g(x^t, y^{t+1})\right]^{-1} \nabla_y f(x^t, y^{t+1})$$

to efficiently approximate the hyper-gradient $\nabla \Phi$ in equation 2.

Let $H_i(x^t, y^{t+1})$ is a hyper-gradient estimator of client $i$ to approximate $\nabla f(x^t, y^{t+1})$ and $F^t := \sigma\{y^0, x^0, ..., y^t, x^t, y^{t+1}\}$ denotes the filtration that captures all the randomness up to the $t$-th outer loop. We denote $\overline{H}_i(x^t, y^{t+1}) := \mathbb{E}[H_i(x^t, y^{t+1})|F^t]$ and $\overline{H}^t := \mathbb{E}[\frac{1}{m}\sum_{i \in \mathcal{M}} \overline{H}_i(x^{t-\tau_i^t}, y^{t-\rho_i^t+1})]$, where $\tau_i^t$ ($\rho_i^t$) is the outer (inner) rounds asynchronous delay of client $i$ in round $t$, respectively.

### 3.2 ALGORITHM DESIGN OF AFBO

The challenge points in solving the bilevel problem in problem 1 lies in computing the federated hyper-gradient $\nabla \Phi(x) = (1/m) \sum_{i=1}^m \nabla f_i(x, y^*(x))$, whose explicit form can be obtained as follows via implicit differentiation.

We can find $\nabla \Phi(x)$ through $\nabla \Phi(x) = (1/m) \sum_{i \in \mathcal{M}} \nabla f_i(x, y^*(x)) = (1/m) \sum_{i \in \mathcal{M}} \nabla_x f_i(x, y^*(x)) - \nabla_{xy}^2 g_i(x, y^*(x)) \times \left[\nabla_{yy}^2 g_i(x, y^*(x))\right]^{-1} \nabla_y f_i(x, y^*(x))$ with Assumption 2, where $\nabla_{yy}^2 g(x, y)$ is defined as the Hessian matrix of $g$ w.r.t. $y$ and $\nabla_{xy}^2 g(x, y)$ is defined as

$$\nabla_{xy}^2 g(x, y) := \begin{bmatrix} \frac{\partial^2}{\partial x_1 \partial y_1} g(x, y) & ... & \frac{\partial^2}{\partial x_1 \partial y_q} g(x, y) \\ & ... & \\ \frac{\partial^2}{\partial x_p \partial y_1} g(x, y) & ... & \frac{\partial^2}{\partial x_p \partial y_q} g(x, y) \end{bmatrix}$$

To calculate the above equation, we need to overcome some difficulties in the federated setting, (i) It is required to approximate the minimizer $y^*(x)$ of the lower-level problem, which can introduce a bias due to the client drift, especially when there are heterogenous local iteration numbers and partial participation; (ii) The computation of a series of global Hessian-vector products lies in a nonlinear manner (i.e., $\frac{1}{M} \sum_{i=1}^M \frac{\partial F_i(x,y)}{\partial y} \frac{\partial y}{\partial x} \neq \frac{\partial F(x,y)}{\partial y} \frac{\partial y}{\partial x}$), which can introduce a large estimation variance; (iii) The federated hyper-gradient estimation may suffer from a bias due to both the upper- and lower-level client drifts including partial participation, asynchronous delay, and unbalanced

computing and/or communication abilities, etc. To address those challenges, we propose a double-loop scheme Anarchic Federated Bilevel Optimization (AFBO) algorithm. For the federated hyper-gradient estimation issue, we use a similar estimation in (Huang et al., 2023), which can be expressed as,

$$H_i(x^t, y^{t+1}) = \nabla_x f_i(x^t, y^{t+1}; \phi_i^t) - \nabla_{xy}^2 g_i(x^t, y^{t+1}; \rho_i^t)$$

$$\times \left[ \frac{N}{l_{g,1}} \prod_{l=1}^{N_i} (I - \frac{1}{l_{g,1}} \nabla_{yy}^2 g_i(x^t, y^{t+1}; \zeta_i^t)) \right] \times \nabla_y f_i(x^t, y^{t+1}; \xi_i^t),$$

where the bias of $H_i$ depends on $N_i$. Then, we aggregate $\nabla\Phi(x)$ through $\nabla\Phi(x^t) = (1/m) \sum_{i \in \mathcal{M}} H_i(x^{t-\tau_i^t}, y^{t-\rho_i^t+1})$, where $\tau_i^t$ is the time index of client $i$'s last communication with the server at round $t$. By using the most recent updates of all clients, the asynchronous bias can be eliminated, which is proved in Theorem 1. In addition, AFBO uses aligned update (i.e., $\frac{1}{E_i^{t,k}} \sum_{c=0}^{E_i^{t,k}-1} \nabla_y g_{i,c}$) to correct the bias from heterogeneous local iteration numbers. In this way, AFBO gives the most flexible to the FBO training process.

---

**Algorithm 1** Anarchic Federated Bilevel Optimization (AFBO)

---

1: **input:** full client index set $\mathcal{M}$, initial point $(x_0, y_0)$, $N \in \mathbb{N}^+$, local computation delays $\{\tau_i^t | i \in \mathcal{M}, t \in [1, T]\}$, the number of local iterations $\{E_i^{t,k} | \forall i \in \mathcal{M}, t \in [0, T-1]\}$
2: **for** Round $t = 0$ to $T - 1$ **do**
3:    **for Client** $i = 1$ to $N$ **do**
4:       **for** $k = 0$ to $K^t - 1$ **do**
5:          **for** $e = 0$ to $E_i^{t,k} - 1$ **do**
6:             **Client** $i$ computes $G_{i,e}^{t,k} = \nabla_y g_i(x^t, y_{i,e}^{t,k}; \xi_{i,e}^{t,k})$;
7:          **end for**
8:          **Client** $i$ sends $G_i^{t,k} = \frac{1}{E_i^{t,k}} \sum_{e=0}^{E_i^{t,k}-1} G_{i,e}^{t,k}$ to the server;
9:          **Server** records the delay of each client, i,e., set $\rho_i^t = 0$ and store $G_i^{t,k}$ on the server for updating clients and $\rho_i^t = \rho_i^{t-1} + 1$ for non-updating clients;
10:         **Server** computes $G^{t,k} = \frac{1}{M} \sum_{i \in M} G_i^{t,k-\rho_i^k}$, $y^{t,k+1} = y^{t,k} - \beta_{t,k} G^{t,k}$, and broadcasts $y^{t,k+1}$;
11:       **end for**
12:       **Client** $i$ computes $H_i^t = APHE(x^t, y^{t+1})$ (note $y^{t+1} = y^{t,K^t-1}$) and sends $H_i^t$ to the server;
13:    **end for**
14:    **Server** records the delay of each client, i,e., set $\tau_i^t = 0$ and store $H_i^t$ on the server for updating clients and $\tau_i^t = \tau_i^{t-1} + 1$ for non-updating clients;
15:    **Server** computes $H^t = \frac{1}{M} \sum_{i \in \mathcal{M}} H_i^{t-\tau_i^t}$, $x^{t+1} = x^t - \eta_t H^t$, and broadcasts $x^{t+1}$;
16: **end for**

---

### 3.3 PROCEDURE OF AFBO

Next, we present the full procedure of AFBO, which is formally described in Algorithm 1. We first define two types of clients in each round $t$: 1) updating clients who have completed their local computations and also communications of their local models to the server in round $t$, so that their local models are used to update the global model in round $t$; 2) non-updating clients who do not participate (i.e., do not perform any local computation or any communication) in round $t$, or who participate but have not completed their local computations or communications of their local models to the server in round $t$.

First of all, for our rounds of AFBO, the server receives the update from updating clients, and retrieves the most recent local models of non-updating clients from the server's memory (since they were lastly updated to the server when the non-updating clients in this round do their communication in previous rounds) to aggregate the global model and broadcasts the latest global model to the updating clients. After clients receive the global model, clients start their inner rounds of training. For

---

**Algorithm 2** Anarchic Parallel hyper-gradient Estimator (APHE)

---

1: **input:** $N = \max_{i \in \mathcal{M}} \{N_i\}$
2: Client $i$ receive $y^{t+1}$ from server;
3: Client $i$ computes $d_i = \nabla_x f_i(x^t, y^{t+1}; \xi_i)$, $W_i = \nabla^2_{xy} g_i(x^t, y^{t+1}; \phi_i)$, and $p_{i,0} = \frac{N}{l_{g,1}} f_i(x^t, y^{t+1}; \theta_i)$;
4: **for** $l = 0$ to $N$ **do**
5:    **if** $l < N_i$ **then**
6:       $p_{i,l} = (I - \frac{1}{l_{g,1}} \nabla^2_{yy} g_i(x^t, y^{t+1}; \zeta_{i,l})) p_{i,l-1}$;
7:    **else**
8:       $p_{i,l} = p_{i,l-1}$;
9:    **end if**
10: **end for**
11: **Return:** $H_i^t = d_i - W_i \times p_{i,N}$.

---

the client $i$ receiving the global model, she uses the federated hyper-gradient estimation method with her own constant $N^i$ to control the Hessian inverse approximation accuracy. After she performs $E_i^{t,k}$ local iterations of inner rounds SGD, she aligns her local updates (i.e., $G_i^{t,k} = \frac{1}{E_i^{t,k}} \sum_{e=0}^{E_i^{t,k}-1} G_{i,e}^{t,k}$), and computes $H_i^t$ using Algorithm 2, then she sends $H_i^t$ to the server to update global model. Clients and the server repeat these processes until the global model converges to a $\epsilon$-error point.

## 4 CONVERGENCE ANALYSIS OF AFBO

### 4.1 DEFINITIONS AND ASSUMPTIONS

Throughout this paper, we make the following assumptions, as typically adopted in federated bilevel optimization.

**Definition 1** *A function $h : \mathbb{R} \to \mathbb{R}$ is Lipschitz continuous with constant $L$, if $\|h(x) - h(y)\| \le L \|x - y\|$, $\forall x, y \in \mathbb{R}$.*

**Definition 2** *A function $h : \mathbb{R} \to \mathbb{R}$ is strongly convex with constant $\mu$, if $h(y) - h(x) \ge \langle \nabla h(x), y - x \rangle + \frac{\mu}{2} \|y - x\|^2$, $\forall x, y \in \mathbb{R}$.*

**Definition 3** *A solution $x$ is $\epsilon$-accurate stationary point if $\mathbb{E}\left[ \|\nabla \Phi(x)\|^2 \right] \le \epsilon$, where $x$ is the output of an algorithm.*

Let $z = (x, y) \in \mathbb{R}^{p+q}$ denote all parameters.

**Assumption 1** *(Lipschitz properties) For all $i \in \mathcal{M}$: $f_i(x)$, $\nabla f_i(z)$, $\nabla g_i(z)$, $\nabla^2 g_i(z)$ are $l_{f,0}$, $l_{f,1}$, $l_{g,1}$, $l_{g,2}$-Lipshitz continuous, respectively.*

**Assumption 2** *(Strong convexity) For all $i \in \mathcal{M}$: $g_i(x, y)$ is $\mu_g$-strongly convex in $y$ for any fixed $x \in \mathcal{N}^q$.*

**Assumption 3** *(Unbiased estimators) For all $i \in \mathcal{M}$: $\nabla f_i(z; \xi)$, $\nabla g_i(z; \zeta)$, $\nabla^2 g_i(z; \zeta)$ are unbiased estimators of $\nabla f_i(z)$, $\nabla g_i(z)$, $\nabla^2 g_i(z)$, respectively.*

**Assumption 4** *(Bounded local SGD variances). For all $i \in \mathcal{M}$: there exist constants $\sigma_f^2$, $\sigma_{g,1}^2$, and $\sigma_{g,2}^2$, such that $\mathbb{E}_\xi\left[ \|\nabla f_i(z; \xi) - \nabla f_i(z)\|^2 \right] \le \sigma_f^2$, $\mathbb{E}_\zeta\left[ \|\nabla g_i(z; \xi) - \nabla g_i(z)\|^2 \right] \le \sigma_{g,1}^2$, $\mathbb{E}_\zeta\left[ \|\nabla^2 g_i(z; \xi) - \nabla^2 g_i(z)\|^2 \right] \le \sigma_{g,2}^2$.*

**Assumption 5** *There exists a constant $\sigma_g$, such that $\mathbb{E}\left[ \|\nabla g_i(z) - \nabla g(z)\|^2 \right] \le \sigma_g^2$, where the expectation $\mathbb{E}$ is taken over the client index $i$.*

**Assumption 6** *There exists a constant $\tau_M$ and $\rho_M$, such that all clients must participate in training at least every $\tau_M$ round for the outer loop and every $\rho_M$ round for the inner loop.*

These assumptions are common in the bilevel optimization literature (Ghadimi & Wang, 2018; Chen et al., 2022a; Ji et al., 2021). Assumption 1 requires that the inner and outer functions are well-behaved. Specifically, strong-convexity of the inner objective is a recurring assumption in bilevel optimization theory implying a unique solution to the inner minimization in equation 1. The Assumption 6 is used in many asynchronous works, such as (Yang et al., 2022b; Wang et al., 2020).

## 4.2 MAIN RESULTS

Next, we present a theoretical performance guarantee for the AFBO algorithm via convergence analysis.

**Theorem 1** *Suppose Assumption 1 to 6 hold, and pick $\eta_t \leq \min\{a_1, \frac{T}{2l_{g,1}a_2}\}$ and $\beta_{t,k} < \frac{a_2\eta_t}{T}$, where $a_1 = \frac{1}{2L_f + 4M_f L_y + \frac{2M_f L_{yx}}{L_y \alpha}}$ and $a_2 = \frac{5M_f L_y}{\mu_g} + \frac{\alpha L_{yx}\hat{D}_f a_1}{2\mu_g}$. Let $W^t \triangleq \Phi(x^t) + \frac{M_f}{L_y}\|y^t - y^*(x^t)\|^2$, and $W^*$ is the optimal point of function $W$. Then, the sequence generated by the AFBO algorithm satisfies*

$$\frac{1}{T}\sum_{t=0}^{T-1}\mathbb{E}\left[\left\|\nabla\Phi(x^t)\right\|^2\right] \leq \frac{2}{\bar{\eta}T}\left[\mathbb{E}[W^0] - \mathbb{E}[W^*]\right] + \frac{2}{\bar{\eta}T}\sum_{t=0}^{T-1}(\frac{M_f U_2}{L_y} + \eta_t M_f^2)\frac{4(\sigma_g^2 + \sigma_{g,1}^2)}{mE^t}\sum_{k=0}^{K^t-1}\beta_{t,k}^2$$

$$+ b^2 + \frac{1}{\bar{\eta}T}\sum_{t=0}^{T-1}(\frac{\eta_t^2 L_f}{m} + \frac{3M_f U_3}{L_y})\hat{\sigma}_f^2 + \frac{1}{\bar{\eta}T}\sum_{t=0}^{T-1}(\frac{3L_f\eta_t^2}{m^2}\sum_{i=1}^{m}\tau_i^t + \frac{12L_f\eta_t^2}{m^2})\hat{D}_f$$

*where $U_2 = 1 + 4M_f L_y\eta_t + \frac{\alpha L_{yx}\hat{D}_f^2\eta_t^2}{2}$, $U_3 = \frac{L_y^2\eta_t^2}{m} + \frac{L_{yx}\eta_t^2}{2\alpha m}$ for any $\alpha > 0$, $b = \kappa_g l_{f,1}(1-\frac{1}{\kappa_g})^N$, $\bar{\eta} = \frac{1}{T}\sum_{t=0}^{T-1}\eta_t$, $\tau_i^t$ is the asynchronous delay of the outer loop and $\frac{1}{E^t} = \frac{1}{m}\sum_{k=0}^{K^t-1}\sum_{i\in\mathcal{M}}\frac{1}{E_i^{t-\tau_i^t, k-\rho_i^k}}$.*

**Remark:** Note that the convergence error bound consists of five parts: a vanishing term that decreases and goes to 0 as the number of rounds $T$ increases, and four non-vanishing (constant) terms that depends on the parameters of the problem instance and is independent of $T$. The first non-vanishing term relates to the total number of participating clients ($m$), the number of local iterations ($E^t$), SGD variances of the lower level ($\sigma_{g,1}$), and the variance of local and global gradients ($\sigma_g$). The second non-vanishing term depends on the Hessian inverse approximation accuracy. The third non-vanishing term depends on SGD variances ($\hat{\sigma}_f$) and the total number of clients ($m$). The last non-vanishing term depends on the upper bound of the estimation of the overall objective gradient ($\hat{D}_f$), the average asynchronous delay ($\frac{1}{m}\sum_{i=1}^{m}\tau_i^t$), and the total number of clients ($m$). The decay rate of the vanishing term matches that of the typical SGD methods.

We observe that the first non-vanishing term involves both the variance of the local and the global gradient in the lower-level ($\sigma_g$) and the lower-level local gradient variance ($\sigma_{g,1}$), and depends on the total number of clients ($m$) and the number of local iterations ($\frac{1}{E^t}$). This error term is due to the variance of stochastic gradients, and it increases as the SGD variance increases and decreases as the total number of clients increases. In addition, the lower non-IID heterogeneity can decrease the error bound of this term. If we run more inner loops, then the error of the first term will decrease. It is also highly affected by the stepsize of the inner loop, which requires us to choose a sufficiently small learning rate. The second term only depends on the rounds of Hessian-vector calculation. The third term is related to the SGD variance of the overall objective ($\hat{\sigma}_f$) and depends on the total number of clients ($m$). The last term of the non-vanishing term involves the bound of the SGD variance of the overall objective ($\hat{D}_f$), the total number of clients ($m$), and the average asynchronous delay ($\frac{1}{m}\sum_{i=1}^{m}\tau_i^t$). The smaller average asynchronous delay decreases the bound of the last term. It can be found that the upper bound of the convergence rate is increasing as the average of outer loop delay for all rounds and all clients increases. Although there are 4 non-vanishing terms, only the last term involves the asynchronous delay (i.e., $\frac{1}{\bar{\eta}T}\sum_{t=0}^{T-1}(\frac{3L_f\eta_t^2}{m^2}\sum_{i=1}^{m}\tau_i^t + \frac{12L_f\eta_t^2}{m^2})\hat{D}_f$). When it turns to a synchronous FBO (i.e., $\tau_i^t = 1$), the convergence bound becomes $\frac{1}{\bar{\eta}T}\sum_{t=0}^{T-1}(\frac{15L_f\eta_t^2}{m})\hat{D}_f$,

which matches previous works. When the asynchronous delay ($\tau_i^t$) increases, we need to choose a smaller step size to keep a similar convergence bound. Intuitively, we need to use a smaller step size to make the delayed gradients affect less global gradients. To make all the non-vanishing terms small, sufficiently small learning rates $\eta_t$ and $\beta_{t,k}$, a large number of inner loops, and a large number of clients should be chosen. Based on Theorem 1, we obtain the following convergence rate for the proposed AFBO algorithm with a proper choice of the learning rate.

**Corollary 1** *Let the stepsize be $\eta_t = \sqrt{\frac{m}{T}}$, $K^t \beta_m^2 \leq \frac{1}{T}$ and $N = \mathcal{O}(logT)$. Then it yields*

$$\frac{1}{T} \sum_{t=0}^{T-1} \mathbb{E}\left[\left\|\nabla f(x^t)\right\|^2\right] \leq \mathcal{O}(\frac{1}{\sqrt{mT}}) + \mathcal{O}(\frac{1}{m\sqrt{mT}}) + \mathcal{O}(\frac{1}{T}) + \mathcal{O}(\frac{1}{m\sqrt{mT}}) + \mathcal{O}(\frac{1}{m\sqrt{mT}}).$$

**Remark:** It has been shown that asynchronous FL algorithms under the non-convex setting can achieve a convergence rate of $\mathcal{O}(1/\sqrt{T})$ (e.g., AsyncCommSGD (Avdiukhin & Kasiviswanathan, 2021), AFA-CD (Yang et al., 2022b)). As our algorithm, which is asynchronous, can reach a convergence rate of $\mathcal{O}(1/\sqrt{T})$, it matches that of the existing synchronous algorithms. In addition, the major term in the upper bound ($\mathcal{O}(1/\sqrt{mT})$, which shows it achieves a linear speedup. In addition, we need $T = \mathcal{O}(\kappa_g^5 \epsilon^{-2}/n)$ to achieve an $\epsilon$-accurate stationary point. Compared with FedNest (Tarzanagh et al., 2022), our complexity has the same dependence on $\kappa$ and $\epsilon$, but a better dependence on $n$ due to the linear speedup.

### 4.3 SKETCH OF PROOF

In this subsection, we highlight the key steps of the proof towards Theorem 1 as well as the differences between our analysis and the existing results. The challenges of our proof mainly lie in obtaining an upper bound for the errors of the asynchronous local gradient and the global gradient.

To prove Theorem 1, we find the difference between two continuous Lyapunov functions, which is

$$\mathbb{E}[W^{t+1}] - \mathbb{E}[W^t] = \Phi(x^{t+1}) - \Phi(x^t) + \frac{M_f}{L_y}(\left\|y^{t+1} - y^*(x^{t+1})\right\|^2 - \left\|y^t - y^*(x^t)\right\|^2). \quad (3)$$

The difference in equation 3 consists of two terms: the first term quantifies the descent of the overall objective function ($\Phi(x)$); the second term characterizes the descent of the lower-level errors ($y^t - y^*(x^t)$ and $y^{t+1} - y^*(x^{t+1})$). Our proof can be summarized as the descent of the overall objective function (Lemma 1), and the upper bound of the lower level problem (Lemma 2).

**Lemma 1** *Suppose Assumptions 1, 2, 3, and 4 hold. We have*

$$\mathbb{E}[\Phi(x^{t+1})] - \mathbb{E}[\Phi(x^t)] \leq (L_f \eta_t^2 - \frac{\eta_t}{2})\mathbb{E}\left[\left\|\overline{H}(x^t, y^{t+1})\right\|^2\right] - \frac{\eta_t}{2}\left\|\nabla\Phi(x^t)\right\|^2 + \eta_t b^2$$
$$+ \eta_t M_f^2 \mathbb{E}\left[\left\|y^{t+1} - y^*(x^t)\right\|^2\right] + \frac{L_f \eta_t^2}{m}(\frac{3}{m}\hat{D}_f \sum_{i=1}^m \tau_i^t + 3\hat{\sigma}_f + \frac{12\hat{D}_f}{m})$$

Lemma 1 shows that the gradient descent of the overall objective function is related to the delay of all clients ($\sum_{i=1}^m \tau_i^t$). It is because we use the most recent updates of all clients to compensate for the bias of asynchronous communication. If we use synchronous communication ($\tau_i^t = 1$), the above result can match the general synchronous BO works.

**Lemma 2** *Suppose Assumptions 1, 2, 3, 4, 5 hold. Letting $\beta_t \leq \frac{1}{2l_{g,1}}$, we have*

$$\mathbb{E}\left[\left\|y^{t+1} - y^*(x^t)\right\|^2\right] \leq \left(\prod_{k=0}^{K^t-1}(1 - \beta_{t,k}\mu_g)\right)\mathbb{E}\left[\left\|y^t - y^*(x^t)\right\|^2\right] + \frac{4(\sigma_g^2 + \sigma_{g,1}^2)}{mE^t}\sum_{k=0}^{K^t-1}\beta_{t,k}^2$$

Lemma 2 shows that the error of the two consecutive inner-loop rounds is smaller when the local iterations ($\frac{1}{E^t} = \frac{1}{m}\sum_{k=0}^{K^t-1}\sum_{i\in\mathcal{M}}\frac{1}{E_i^{t-\tau_i^t, k-\rho_i^k}}$) increase. This is because we allow clients to do multiple local iterations and align their updates through $G_i^{t,k} = \frac{1}{E_i^{t,k}}\sum_{e=0}^{E_i^{t,k}-1}G_{i,e}^{t,k}$.

## 5 NUMERICAL EXPERIMENTS

In this section, we conduct experiments on hyper-parameter optimization tasks (i.e., the data hyper-cleaning task and regularization coefficient optimization task), which are important problems in multi-task machine learning, in the distributed setting to evaluate the performance of the proposed AFBO and validate our theoretical results. The experiments exactly follow the same settings described by a recent FBO, ADBO ((Jiao et al., 2023)). To make our paper self-contained, we defer the detailed setting of experiments to the appendix (Appendix B) due to space limitations.

### 5.1 DATA HYPER-CLEANING TASK

Following (Ji et al., 2021; Yang et al., 2021), the proposed AFBO algorithm is compared with ADBO (Jiao et al., 2023) and distributed bilevel optimization method FedNest (Tarzanagh et al., 2022) on the distributed data hyper-cleaning task (Chen et al., 2022b) on MNIST datasets (LeCun et al., 1998). Data hyper-cleaning (Chen et al., 2022b) involves training a classifier in a contaminated environment where each training data label is changed to a random class number with a probability (i.e., the corruption rate). The test accuracy for 4 algoirithms with IID and non-IID datasets are shown in Fig. 1 and Fig. 2. We can observe that the proposed AFBO is the most efficient algorithm since the asynchronous setting is considered in AFBO, the server can update its variables once it receives updates from updating clients and it allows multiple local iterations which makes fully use of clients' computing resources.

### 5.2 REGULARIZATION COEFFICIENT OPTIMIZATION TASK

Following (Chen et al., 2022a), we compare the performance of AFBO with baseline algorithms FedNest, SDBO, and ADBO on the regularization coefficient optimization task using Covertype datasets. The results on Covertype dataset are shown in Fig. 3, which shows AFBO is the best algorithm among them. Next, we assume there are at most five stragglers in the distributed system, and the mean of (communication + computation) delay of stragglers is five times the delay of normal clients. The result is shown in Fig. 4. It is found that the efficiency of the synchronous distributed algorithms (FedNest and SDBO) will be significantly affected, while the proposed AFBO and ADBO suffer slightly from the straggler problem since it is an asynchronous method and only considers updating clients.

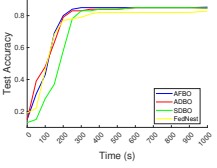 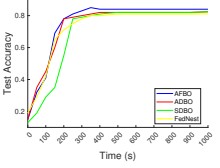 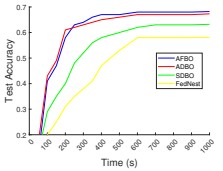 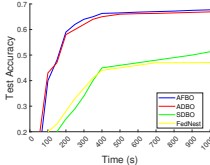

| Figure 1: Test accuracy vs time on IID MINIST. | Figure 2: Test accuracy vs time on non-IID MINIST. | Figure 3: Test accuracy vs time on Covertype. | Figure 4: Test accuracy vs time on Covertype. |

## 6 CONCLUSION

In this paper, we proposed a double-loop scheme Anarchic Federated Bilevel Optimization (AFBO) algorithm, which enables clients to flexibly participate in federated bilevel optimization training according to their heterogeneous and time-varying computation and communication capabilities, and also efficiently by improving utilization of their computation and communication resources. We provided a convergence analysis, which shows that the performance of AFBO matches that of the existing benchmarks. We have also conducted simulations using real-world datasets for FBO benchmarks to demonstrate the efficiency of AFBO. For future work, we will explore AFBO in other settings of FBO, such as decentralized or hierarchical networks of clients. These cases will be more challenging to study due to the complex communication structure.

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
