APPENDIX FOR ANARCHIC FEDERATED BILEVEL OPTIMIZATION

## A  PROOF OF THEOREMS AND LEMMAS

We first cite some useful Lemmas in the previous papers, then we provide the required Lemmas and its proof in our paper.

### A.1  USEFUL LEMMAS FROM PREVIOUS RESULTS

**Lemma 3 ((Ghadimi & Wang, 2018) Lemma 2.2)** *Under Assumptions 1 and 2, we have*

$$\|\nabla \Phi(x_1) - \nabla \Phi(x_2)\| \leq L_f \|x_1 - x_2\|,$$
$$\|y^*(x_1) - y^*(x_2)\| \leq L_y \|x_1 - x_2\|,$$

*where* $L_f := l_{f,1} + \frac{l_{g,1}(l_{f,1}+M_f)}{\mu_g} + \frac{l_{f,0}(l_{g,2}+\frac{l_{g,1}l_{g,2}}{\mu_g})}{\mu_g}$ *and* $L_y = \frac{l_{g,1}}{\mu_g}$.

*For all* $i \in \mathcal{M}$, *we have*

$$\|\overline{\nabla} f_i(x_1, y) - \overline{\nabla} f_i(x_1, y^*(x_1))\| \leq M_f \|y - y^*(x_1)\|,$$
$$\|\overline{\nabla} f_i(x_1, y) - \overline{\nabla} f_i(x_2, y)\| \leq M_f \|x_1 - x_2\|,$$

*where* $\overline{\nabla} f_i(x, y) := \nabla_x f_i(x, y) - \nabla^2_{xy} g(x, y)[\nabla^2_{yy} g(x, y)]^{-1} \nabla_y f_i(x, y)$ *and* $M_f := l_{f,1} + \frac{l_{g,1}l_{f,1}}{\mu_g} + \frac{l_{f,0}}{\mu_g}(l_{g,2} + \frac{l_{g,1}l_{g,2}}{\mu_g})$

*Proof.* The proof is similar to (Ghadimi & Wang, 2018), Lemma 2.2.

**Lemma 4 ( (Chen et al., 2021) Lemma 2)** *Under Assumptions 1, 2, and 3, we have*

$$\|\nabla y^*(x_1) - \nabla y^*(x_2)\| \leq L_{yx} \|x_1 - x_2\|,$$

*where* $L_{yx} := \frac{l_{g,2}(1+L_y)}{\mu_g} + \frac{l_{g,1}l_{g,2}}{\mu_g^2}(1 + L_y)$.

*Proof.* The proof is similar to Lemma 2 of (Chen et al., 2021).

**Lemma 5** *(Huang et al., 2023) Under Assumption 2, we have*

$$\nabla \Phi(x) = \nabla f(x, y^*(x)) = \nabla_x f(x, y^*(x)) - \nabla^2_{xy} g(x, y^*(x)) \times \left[\nabla^2_{yy} g(x, y^*(x))\right]^{-1} \nabla_y f(x, y*(x)), \quad (4)$$

*where* $\nabla^2_{yy} g(x, y)$ *is defined as the Hessian matrix of* $g$ *w.r.t* $y$ *and* $\nabla^2_{xy} g(x, y)$ *is defined as*

$$\nabla^2_{xy} g(x, y) := \begin{bmatrix} \frac{\partial^2}{\partial x_1 \partial y_1} g(x, y) & \dots & \frac{\partial^2}{\partial x_1 \partial y_q} g(x, y) \\ & \dots & \\ \frac{\partial^2}{\partial x_p \partial y_1} g(x, y) & \dots & \frac{\partial^2}{\partial x_p \partial y_q} g(x, y) \end{bmatrix}$$

*Proof.* The proof is similar to (Huang et al., 2023) Lemma 3.

**Lemma 6** *((Hong et al., 2020) Lemma 1 and (Chen et al., 2021) Lemma 2) Suppose Assumptions 1, 2 3, and 4 hold, we can get*

$$\mathbb{E}\left[\left\|H_i(x^t, y^{t+1}) - \overline{H}_i(x^t, y^{t+1})\right\|^2\right] \leq \hat{\sigma}_f$$

$$\mathbb{E}\left[\left\|H_i(x^t, y^{t+1})\right\|^2 |F^t\right] \leq \hat{D}_f$$

*where* $\hat{\sigma}_f = \sigma_f^2 + \frac{3}{\mu^2}((\sigma_f^2 + l_{f,0}^2)(\sigma_{g,2}^2 + 2l_{g,1}^2) + \sigma_f^2 l_{g,1}^2)$ *and* $\hat{D}_f = (l_{f,0} + \frac{2l_{g,1}l_{f,1}}{\mu_g})^2 + \hat{\sigma}_f$.

The proof is similar to (Hong et al., 2020) Lemma 1 and (Chen et al., 2021) Lemma 2.

**Lemma 7** *((Tarzanagh et al., 2022), Lemma 2.2) Suppose Assumptions 1, 2 3, and 4 hold, we can get*

$$\mathbb{E}\left[\left\|\overline{H}(x^t, y^{t+1}) - \overline{\nabla}f(x^t, y^{t+1})\right\|^2\right] \leq b^2$$

*where $b = \kappa_g l_{f,1}(1 - \frac{1}{\kappa_g})^N$, $\kappa_g := \frac{L_g}{\mu_g}$, and $N$ is the input parameter of Algorithm 1.*

*Proof.* The proof is similar to Lemma 2 of (Tarzanagh et al., 2022).

## A.2 PROPOSITION 1 AND ITS PROOF

**Proposition 1** *Suppose Assumptions 1, 2, 3, and 4 hold, we can get*

$$\mathbb{E}\left[\left\|\sum_{i=1}^m \overline{H}_i(x^t, y^{t+1}) - \overline{H}(x^t, y^{t+1})\right\|^2\right] \leq 4m\hat{D}_f$$

*Proof*:

$$\mathbb{E}\left[\left\|\sum_{i=1}^m \overline{H}_i(x^t, y^{t+1}) - \overline{H}(x^t, y^{t+1})\right\|^2\right]$$

$$=\mathbb{E}\left[\left\|\sum_{i=1}^m \overline{H}_i(x^t, y^{t+1}) - \sum_{i=1}^m \overline{H}_i(x^{t-\tau_i^t}, y^{t-\rho_i^t+1})\right\|^2\right]$$

$$=\mathbb{E}\left[\left\|\sum_{i=1}^m \left[\overline{H}_i(x^t, y^{t+1}) - \overline{H}_i(x^{t-\tau_i^t}, y^{t-\rho_i^t+1})\right]\right\|^2\right]$$

$$\leq m\sum_{i=1}^m \mathbb{E}\left[\left\|\overline{H}_i(x^t, y^{t+1}) - \overline{H}_i(x^{t-\tau_i^t}, y^{t-\rho_i^t+1})\right\|^2\right]$$

$$\leq 2m\sum_{i=1}^m \left\{\mathbb{E}\left[\left\|\overline{H}_i(x^t, y^{t+1})\right\|^2\right] + \mathbb{E}\left[\left\|\overline{H}_i(x^{t-\tau_i^t}, y^{t-\rho_i^t+1})\right\|^2\right]\right\}$$

$$\leq 4m\hat{D}_f,$$

where we use the Lemma 6 for the last inequality.

## A.3 PROPOSITION 2 AND ITS PROOF

**Proposition 2** *Suppose Assumptions 1, 2, 3, and 4 hold, we can get*

$$\mathbb{E}\left[\left\|\frac{1}{m}\sum_{i=1}^m H_i(x^{t-\tau_i^t}, y^{t-\rho_i^t+1}) - \overline{H}(x^t, y^{t+1})\right\|^2\right] \leq \frac{3}{m}\hat{D}_f\sum_{i=1}^m \tau_i^t + 3\hat{\sigma}_f + \frac{12\hat{D}_f}{m}$$

*where $\hat{\sigma}_f = \sigma_f^2 + \frac{3}{\mu^2}((\sigma_f^2 + l_{f,0}^2)(\sigma_{g,2}^2 + 2l_{g,1}^2) + \sigma_f^2 l_{g,1}^2)$*

*Proof*:

$$\mathbb{E}\left[\left\|\frac{1}{m}\sum_{i=1}^m H_i(x^{t-\tau_i^t}, y^{t-\tau_i^t+1}) - \overline{H}(x^t, y^{t+1})\right\|^2\right]$$

$$\leq\mathbb{E}\left[\left\|\frac{1}{m}\sum_{i=1}^m \left(H_i(x^{t-\tau_i^t}, y^{t-\tau_i^t+1}) - H_i(x^t, y^{t+1}) + H_i(x^t, y^{t+1}) - \overline{H}_i(x^t, y^{t+1}) + \overline{H}_i(x^t, y^{t+1})\right) - \overline{H}(x^t, y^{t+1})\right\|^2\right]$$

$$\leq 3\mathbb{E}\left[\left\|\frac{1}{m}\sum_{i=1}^m \left(H_i(x^{t-\tau_i^t}, y^{t-\tau_i^t+1}) - H_i(x^t, y^{t+1})\right)\right\|^2\right] + 3\mathbb{E}\left[\left\|\frac{1}{m}\sum_{i=1}^m \left(H_i(x^t, y^{t+1}) - \overline{H}_i(x^t, y^{t+1})\right)\right\|^2\right]$$

$$+ 3\mathbb{E}\left[\left\|\frac{1}{m}\sum_{i=1}^{m}\overline{H}_i(x^t, y^{t+1}) - \overline{H}(x^t, y^{t+1})\right\|^2\right]$$

$$\leq \frac{3}{m}\sum_{i=1}^{m}\mathbb{E}\left[\left\|H_i(x^{t-\tau_i^t}, y^{t-\tau_i^t+1}) - H_i(x^t, y^{t+1})\right\|^2\right] + \frac{3}{m}\sum_{i=1}^{m}\mathbb{E}\left[\left\|H_i(x^t, y^{t+1}) - \overline{H}_i(x^t, y^{t+1})\right\|^2\right]$$

$$+ 3\mathbb{E}\left[\left\|\frac{1}{m}\sum_{i=1}^{m}\left(\overline{H}_i(x^t, y^{t+1}) - \overline{H}_i(x^{t-\tau_i^t}, y^{t-\tau_i^t+1})\right)\right\|^2\right]$$

$$\leq \frac{3}{m}\sum_{i=1}^{m}\mathbb{E}\left[\left\|H_i(x^{t-\tau_i^t}, y^{t-\tau_i^t+1}) - H_i(x^t, y^{t+1})\right\|^2\right] + \frac{3}{m}\sum_{i=1}^{m}\mathbb{E}\left[\left\|H_i(x^t, y^{t+1}) - \overline{H}_i(x^t, y^{t+1})\right\|^2\right] + \frac{12\hat{D}_f}{m}$$

$$\leq \frac{3}{m}\hat{D}_f\sum_{i=1}^{m}\tau_i^t + 3\hat{\sigma}_f + \frac{12\hat{D}_f}{m}$$

### A.4  PROOF OF LEMMA 1

*Proof*:

$$\mathbb{E}[\Phi(x^{t+1})] - \mathbb{E}[\Phi(x^t)]$$

$$\leq \mathbb{E}\left[\langle x^{t+1} - x^t, \nabla\Phi(x^t)\rangle\right] + \frac{L_f}{2}\mathbb{E}\left[\left\|x^{t+1} - x^t\right\|^2\right]$$

$$= -\mathbb{E}\left[\left\langle\frac{1}{m}\sum_{i=1}^{m}\eta_t H_i(x^{t-\tau_i^t}, y^{t-\tau_i^t+1}), \nabla\Phi(x^t)\right\rangle\right] + \frac{L_f}{2}\mathbb{E}\left[\left\|\frac{1}{m}\sum_{i=1}^{m}\eta_t H_i(x^{t-\tau_i^t}, y^{t-\tau_i^t+1})\right\|^2\right]$$

Then, for $-\mathbb{E}\left[\left\langle\frac{1}{m}\sum_{i=1}^{m}\eta_t H_i(x^{t-\tau_i^t}, y^{t-\tau_i^t+1}), \nabla\Phi(x^t)\right\rangle\right]$. we have

$$-\mathbb{E}\left[\left\langle\frac{1}{m}\sum_{i=1}^{m}\eta_t H_i(x^{t-\tau_i^t}, y^{t-\tau_i^t+1}), \nabla\Phi(x^t)\right\rangle\right]$$

$$= -\mathbb{E}\left[\eta_t\mathbb{E}\left[\left\langle\frac{1}{m}\sum_{i=1}^{m}H_i(x^{t-\tau_i^t}, y^{t-\tau_i^t+1}), \nabla\Phi(x^t)\right\rangle|F^t\right]\right]$$

$$= -\mathbb{E}\left[\langle\eta_t\overline{H}(x^t, y^{t+1}), \nabla\Phi(x^t)\rangle\right]$$

$$= -\frac{\eta_t}{2}\mathbb{E}\left[\left\|\overline{H}(x^t, y^{t+1})\right\|^2\right] - \frac{\eta_t}{2}\left\|\nabla\Phi(x^t)\right\|^2 + \frac{\eta_t}{2}\mathbb{E}\left[\left\|\overline{H}(x^t, y^{t+1}) - \nabla\Phi(x^t)\right\|^2\right]$$

$$= -\frac{\eta_t}{2}\mathbb{E}\left[\left\|\overline{H}(x^t, y^{t+1})\right\|^2\right] - \frac{\eta_t}{2}\left\|\nabla\Phi(x^t)\right\|^2 + \frac{\eta_t}{2}\mathbb{E}\left[\left\|\overline{H}(x^t, y^{t+1}) - \overline{\nabla}f(x^t, y^{t+1}) + \overline{\nabla}f(x^t, y^{t+1}) - \nabla\Phi(x^t)\right\|^2\right]$$

$$\leq -\frac{\eta_t}{2}\mathbb{E}\left[\left\|\overline{H}(x^t, y^{t+1})\right\|^2\right] - \frac{\eta_t}{2}\left\|\nabla\Phi(x^t)\right\|^2 + \eta_t\mathbb{E}\left[\left\|\overline{H}(x^t, y^{t+1}) - \overline{\nabla}f(x^t, y^{t+1})\right\|^2\right] + \eta_t\mathbb{E}\left[\left\|\overline{\nabla}f(x^t, y^{t+1}) - \nabla\Phi(x^t)\right\|^2\right]$$

$$\leq -\frac{\eta_t}{2}\mathbb{E}\left[\left\|\overline{H}(x^t, y^{t+1})\right\|^2\right] - \frac{\eta_t}{2}\left\|\nabla\Phi(x^t)\right\|^2 + \eta_t b^2 + \eta_t M_f^2\mathbb{E}\left[\left\|y^{t+1} - y^*(x^t)\right\|^2\right]$$

Next, for $\frac{L_f}{2}\mathbb{E}\left[\left\|\frac{1}{m}\sum_{i=1}^{m}\eta_t H_i(x^{t-\tau_i^t}, y^{t-\tau_i^t+1})\right\|^2\right]$, we have

$$\frac{L_f}{2}\mathbb{E}\left[\left\|\frac{1}{m}\sum_{i=1}^{m}\eta_t H_i(x^{t-\tau_i^t}, y^{t-\tau_i^t+1})\right\|^2\right]$$

$$= \frac{L_f\eta_t^2}{2}\mathbb{E}\left[\left\|\frac{1}{m}\sum_{i=1}^{m}\left(H_i(x^{t-\tau_i^t}, y^{t-\tau_i^t+1}) - \overline{H}(x^t, y^{t+1}) + \overline{H}(x^t, y^{t+1})\right)\right\|^2\right]$$

$$\leq L_f \eta_t^2 \mathbb{E}\left[\left\|\frac{1}{m}\sum_{i=1}^m \left(H_i(x^{t-\tau_i^t}, y^{t-\tau_i^t+1}) - \overline{H}(x^t, y^{t+1})\right)\right\|^2\right] + L_f \eta_t^2 \mathbb{E}\left[\left\|\frac{1}{m}\sum_{i=1}^m \overline{H}(x^t, y^{t+1})\right\|^2\right]$$

$$\leq \frac{L_f \eta_t^2}{m}\left(\frac{3}{m}\hat{D}_f \sum_{i=1}^m \tau_i^t + 3\hat{\sigma}_f + \frac{12\hat{D}_f}{m}\right) + L_f \eta_t^2 \mathbb{E}\left[\left\|\overline{H}(x^t, y^{t+1})\right\|^2\right]$$

Combining the above inequalities yields

$$\mathbb{E}[\Phi(x^{t+1})] - \mathbb{E}[\Phi(x^t)] \leq (L_f \eta_t^2 - \frac{\eta_t}{2})\mathbb{E}\left[\left\|\overline{H}(x^t, y^{t+1})\right\|^2\right] - \frac{\eta_t}{2}\left\|\nabla\Phi(x^t)\right\|^2 + \eta_t M_f^2 \mathbb{E}\left[\left\|y^{t+1} - y^*(x^t)\right\|^2\right]$$

$$+ \frac{L_f \eta_t^2}{m}\left(\frac{3}{m}\hat{D}_f \sum_{i=1}^m \tau_i^t + 3\hat{\sigma}_f + \frac{12\hat{D}_f}{m}\right) + \eta_t b^2$$

## A.5 LEMMA 8 AND ITS PROOF

**Lemma 8** *Let $\frac{1}{E^{t,k}} = \frac{1}{m}\sum_{i\in\mathcal{M}}\frac{1}{E_i^{t-\tau_i^t,k}}$, we have*

$$\mathbb{E}\left[\left\|\frac{1}{m}\sum_{i=1}^m \frac{1}{E_i^{t-\tau_i^t,k-\rho_i^k}}\sum_{c=0}^{E_i^{t-\tau_i^t,k-\rho_i^k}-1} \nabla_y g_i(x^{t-\tau_i^t}, y^*(x^{t-\tau_i^t}); \xi_{i,c}^{t-\tau_i^t,k-\rho_i^k})\right\|^2\right] \leq \frac{2\sigma_{g,1}^2 + 2\sigma_g^2}{mE^{t,k}}$$

*Proof:*

$$\mathbb{E}\left[\left\|\frac{1}{m}\sum_{i=1}^m \frac{1}{E_i^{t-\tau_i^t,k-\rho_i^k}}\sum_{c=0}^{E_i^{t-\tau_i^t,k-\rho_i^k}-1} \nabla_y g_i(x^{t-\tau_i^t}, y^*(x^{t-\tau_i^t}); \xi_{i,c}^{t-\tau_i^t,k-\rho_i^k})\right\|^2\right]$$

$$= \frac{1}{m^2}\sum_{i=1}^m \frac{1}{(E_i^{t-\tau_i^t,k-\rho_i^k})^2}\sum_{c=0}^{E_i^{t-\tau_i^t,k-\rho_i^k}-1} \mathbb{E}\left[\left\|\nabla_y g_i(x^{t-\tau_i^t}, y^*(x^{t-\tau_i^t}); \xi_{i,c}^{t-\tau_i^t,k-\rho_i^k})\right\|^2\right] + \frac{1}{m^2}$$

$$\sum_{1\leq i\neq j\leq m}\left\langle \frac{1}{E_i^{t-\tau_i^t,k-\rho_i^k}}\sum_{c=0}^{E_i^{t-\tau_i^t,k-\rho_i^k}-1} \nabla_y g_i(x^{t-\tau_i^t}, y^*(x^{t-\tau_i^t}); \xi_{i,c}^{t-\tau_i^t,k-\rho_i^k}), \right.$$

$$\left. \frac{1}{E_j^{t-\tau_j^t,k-\rho_i^k}}\sum_{c=0}^{E_j^{t-\tau_j^t,k-\rho_i^k}-1} \nabla_y g_j(x^{t-\tau_j^t}, y^*(x^{t-\tau_j^t}); \xi_{j,c}^{t-\tau_j^t,k-\rho_i^k})\right\rangle$$

$$= \frac{1}{m^2}\sum_{i=1}^m \frac{1}{(E_i^{t-\tau_i^t,k-\rho_i^k})^2}\sum_{c=0}^{E_i^{t-\tau_i^t,k-\rho_i^k}-1} \mathbb{E}\left[\left\|\nabla_y g_i(x^{t-\tau_i^t}, y^*(x^{t-\tau_i^t}); \xi_{i,c}^{t-\tau_i^t,k-\rho_i^k})\right\|^2\right]$$

$$\leq \frac{2}{m^2}\sum_{i=1}^m \frac{1}{(E_i^{t-\tau_i^t,k-\rho_i^k})^2}\sum_{c=0}^{E_i^{t-\tau_i^t,k-\rho_i^k}-1} \mathbb{E}\left[\left\|\nabla_y g_i(x^{t-\tau_i^t}, y^*(x^{t-\tau_i^t}); \xi_{i,c}^{t-\tau_i^t,k-\rho_i^k}) - \nabla_y g_i(x^{t-\tau_i^t}, y^*(x^{t-\tau_i^t}))\right\|^2\right]$$

$$+ \frac{2}{m^2}\sum_{i=1}^m \frac{1}{(E_i^{t-\tau_i^t,k-\rho_i^k})^2}\sum_{c=0}^{E_i^{t-\tau_i^t,k-\rho_i^k}-1} \mathbb{E}\left[\left\|\nabla_y g_i(x^{t-\tau_i^t}, y^*(x^{t-\tau_i^t})) - \nabla_y g(x^{t-\tau_i^t}, y^*(x^{t-\tau_i^t}))\right\|^2\right]$$

$$\leq \frac{2\sigma_{g,1}^2 + 2\sigma_g^2}{mE^{t,k}}$$

## A.6 PROOF OF LEMMA 2

*Proof:*

We first show that $\mathbb{E}\left[\left\|y^{t,k+1} - y^*(x^t)\right\|^2\right] \leq (1 - \beta_{t,k}\mu_g)\mathbb{E}\left[\left\|y^{t,k} - y^*(x^t)\right\|^2\right] + \frac{2\beta_{t,k}^2\sigma_g^2}{mE^{t,k}}$

$$\mathbb{E}\left[\left\|y^{t,k+1} - y^*(x^t)\right\|^2\right]$$

$$=\mathbb{E}\left[\left\|y^{t,k} - \frac{\beta_{t,k}}{m}\sum_{i\in\mathcal{M}} G_i^{t-\tau_i^t, k-\rho_i^k} - y^*(x^t)\right\|^2\right]$$

$$=\mathbb{E}\left[\left\|y^{t,k} - y^*(x^t)\right\|^2\right] - \beta_{t,k}\mathbb{E}\left[\left\langle\frac{1}{m}\sum_{i\in\mathcal{M}} G_i^{t-\tau_i^t, k-\rho_i^k}, y^{t,k} - y^*(x^t)\right\rangle\right] + \beta_{t,k}^2\mathbb{E}\left[\left\|\frac{1}{m}\sum_{i\in\mathcal{M}} G_i^{t-\tau_i^t, k-\rho_i^k}\right\|^2\right]$$

$$=\mathbb{E}\left[\left\|y^{t,k} - y^*(x^t)\right\|^2\right] - \beta_{t,k}\mathbb{E}\left[\left\langle\nabla_y g(x^t, y^{t,k}), y^{t,k} - y^*(x^t)\right\rangle\right] + \beta_{t,k}^2\mathbb{E}\left[\left\|\frac{1}{m}\sum_{i\in\mathcal{M}} G_i^{t-\tau_i^t, k-\rho_i^k}\right\|^2\right]$$

$$\leq(1 - \beta_{t,k}\mu_g)\mathbb{E}\left[\left\|y^{t,k} - y^*(x^t)\right\|^2\right] - 2\beta_{t,k}\mathbb{E}\left[g(x^t, y^{t,k}) - g(x^t, y^*(x^t))\right] + \beta_{t,k}^2\mathbb{E}\left[\left\|\frac{1}{m}\sum_{i\in\mathcal{M}} G_i^{t-\tau_i^t, k-\rho_i^k}\right\|^2\right],$$

where $G_i^{t,k} = \frac{1}{E_i^{t,k}}\sum_{c=0}^{E_i^{t,k}-1}\nabla_y g_i(x^t, y^{t,k}; \xi_{i,E_i^{t,k}}^{t,k})$ as defined in Algorithm 1. We use the fact that $G_i^{t,k}$ is an unbiased gradient estimator in the third equality and employ the $\mu_g$-strong convexity of $g(x,y)$ w.t.t. $y$ in the last inequality.

$$\mathbb{E}\left[\left\|\frac{1}{m}\sum_{i\in\mathcal{M}} G_i^{t-\tau_i^t, k-\rho_i^k}\right\|^2\right]$$

$$=\mathbb{E}\left[\left\|\frac{1}{m}\sum_{i=1}^m\frac{1}{E_i^{t-\tau_i^t,k-\rho_i^k}}\sum_{c=0}^{E_i^{t-\tau_i^t,k-\rho_i^k}-1}\nabla_y g_i(x^{t-\tau_i^t}, y^{t-\tau_i^t,k-\rho_i^k}; \xi_{i,E_i^{t-\tau_i^t,k-\rho_i^k}}^{t-\tau_i^t,k-\rho_i^k})\right\|^2\right]$$

$$=\mathbb{E}\left[\left\|\frac{1}{m}\sum_{i=1}^m\frac{1}{E_i^{t-\tau_i^t,k-\rho_i^k}}\sum_{c=0}^{E_i^{t-\tau_i^t,k-\rho_i^k}-1}\left(\nabla_y g_i(x^{t-\tau_i^t}, y^{t-\tau_i^t,k-\rho_i^k}; \xi_{i,E_i^{t-\tau_i^t,k-\rho_i^k}}^{t-\tau_i^t,k-\rho_i^k})\right.\right.\right.$$
$$\left.\left.\left. -\nabla_y g_i(x^{t-\tau_i^t}, y^*(x^{t-\tau_i^t}); \xi_{i,E_i^{t-\tau_i^t,k-\rho_i^k}}^{t-\tau_i^t,k-\rho_i^k}) + \nabla_y g_i(x^{t-\tau_i^t}, y^*(x^{t-\tau_i^t}); \xi_{i,E_i^{t-\tau_i^t,k-\rho_i^k}}^{t-\tau_i^t,k-\rho_i^k})\right)\right\|^2\right]$$

$$=2\mathbb{E}\left[\left\|\frac{1}{m}\sum_{i=1}^m\frac{1}{E_i^{t-\tau_i^t,k-\rho_i^k}}\sum_{c=0}^{E_i^{t-\tau_i^t,k-\rho_i^k}-1}\left(\nabla_y g_i(x^{t-\tau_i^t}, y^{t-\tau_i^t,k-\rho_i^k}; \xi_{i,E_i^{t-\tau_i^t,k-\rho_i^k}}^{t-\tau_i^t,k-\rho_i^k})\right.\right.\right.$$
$$\left.\left.\left. -\nabla_y g_i(x^{t-\tau_i^t}, y^*(x^{t-\tau_i^t}); \xi_{i,E_i^{t-\tau_i^t,k-\rho_i^k}}^{t-\tau_i^t,k-\rho_i^k})\right)\right\|^2\right]$$

$$+ 2\mathbb{E}\left[\left\|\frac{1}{m}\sum_{i=1}^m\frac{1}{E_i^{t-\tau_i^t,k-\rho_i^k}}\sum_{c=0}^{E_i^{t-\tau_i^t,k-\rho_i^k}-1}\nabla_y g_i(x^{t-\tau_i^t}, y^*(x^{t-\tau_i^t}); \xi_{i,E_i^{t-\tau_i^t,k-\rho_i^k}}^{t-\tau_i^t,k-\rho_i^k})\right\|^2\right]$$

$$\overset{7.1}{\leq}\frac{2}{m}\sum_{i=1}^m\frac{1}{E_i^{t-\tau_i^t,k-\rho_i^k}}\sum_{c=0}^{E_i^{t-\tau_i^t,k-\rho_i^k}-1}\mathbb{E}\left[\left\|\nabla_y g_i(x^{t-\tau_i^t}, y^{t-\tau_i^t,k-\rho_i^k}; \xi_{i,E_i^{t-\tau_i^t,k-\rho_i^k}}^{t-\tau_i^t,k-\rho_i^k})\right.\right.$$
$$\left.\left. -\nabla_y g_i(x^{t-\tau_i^t}, y^*(x^{t-\tau_i^t}); \xi_{i,E_i^{t-\tau_i^t,k-\rho_i^k}}^{t-\tau_i^t,k-\rho_i^k})\right\|^2\right] + \frac{4(\sigma_g^2 + \sigma_{g,1}^2)}{mE^{t,k}}$$

$$\overset{7.2}{\leq} \frac{4l_{g,1}}{m} \sum_{i=1}^{m} \frac{1}{E_i^{t-\tau_i^t,k-\rho_i^k}} \sum_{c=0}^{E_i^{t-\tau_i^t,k-\rho_i^k}-1} \mathbb{E}\left[ g_i(x^{t-\tau_i^t}, y^{t-\tau_i^t,k-\rho_i^k}; \xi_{i,E_i^{t-\tau_i^t,k-\rho_i^k}}^{t-\tau_i^t,k-\rho_i^k}) - g_i(x^{t-\tau_i^t}, y^*(x^{t-\tau_i^t}); \xi_{i,E_i^{t-\tau_i^t,k-\rho_i^k}}^{t-\tau_i^t,k-\rho_i^k}) \right.$$

$$\left. - \left\langle \nabla_y g_i(x^{t-\tau_i^t}, y^*(x^{t-\tau_i^t}); \xi_{i,E_i^{t-\tau_i^t,k-\rho_i^k},s}^{t-\tau_i^t,k-\rho_i^k}), y^{t-\tau_i^t,k-\rho_i^k} - y^*(x^{t-\tau_i^t}) \right\rangle \right] + \frac{4(\sigma_g^2 + \sigma_{g,1}^2)}{mE^{t,k}}$$

$$= 4l_{g,1}\mathbb{E}\left[ g(x^t, y^t) - g(x^t, y^*(x^t)) \right] + \frac{4(\sigma_g^2 + \sigma_{g,1}^2)}{mE^{t,k}}$$

where 7.1 uses Lemma 8 and 7.2 uses Lemma 1 in (Woodworth et al., 2020). Enforcing $\beta_{t,k} \leq \frac{1}{l_{g,1}}$ yield

$$\mathbb{E}\left[ \left\| y^{t,k+1} - y^*(x^t) \right\|^2 \right]$$

$$\leq (1 - \beta_{t,k}\mu_g)\mathbb{E}\left[ \left\| y^{t,k} - y^*(x^t) \right\|^2 \right] + 2\beta_{t,k}(2\beta_{t,k}l_{g,1} - 1)\mathbb{E}\left[ g(x^t, y^{t,k}) - g(x^t, y^*(x^t)) \right] + \frac{4(\beta^{t,k})^2(\sigma_g^2 + \sigma_{g,1}^2)}{mE^{t,k}}$$

$$\leq (1 - \beta_{t,k}\mu_g)\mathbb{E}\left[ \left\| y^{t,k} - y^*(x^t) \right\|^2 \right] + \frac{4(\beta^{t,k})^2(\sigma_g^2 + \sigma_{g,1}^2)}{mE^{t,k}}$$

Summing $k = 0$ to $K_m - 1$ and let $\frac{1}{E^t} = \sum_{k=0}^{K_m-1} E^{t,k}$, we have

$$\mathbb{E}\left[ \left\| y^{t,K_m-1} - y^*(x^t) \right\|^2 \right] \leq \left( \prod_{k=0}^{K_m-1} (1 - \beta_{t,k}\mu_g) \right)\mathbb{E}\left[ \left\| y^{t,0} - y^*(x^t) \right\|^2 \right] + \frac{4(\sigma_g^2 + \sigma_{g,1}^2)}{mE^t} \sum_{k=0}^{K_m-1} \beta_{t,k}^2$$

### A.7   Lemma 9 and its proof

**Lemma 9** *Suppose Assumptions 1, 2, 3, and 4 hold, Algorithm 1 guarantees:*

$$\mathbb{E}\left[ \left\| y^{t+1} - y^*(x^{t+1}) \right\|^2 \right] \leq U_1 \mathbb{E}\left[ \left\| \overline{H}(x^t, y^{t+1}) \right\|^2 \right] + U_2 \mathbb{E}\left[ \left\| y^{t+1} - y^*(x^t) \right\|^2 \right] + U_3 \hat{\sigma}_f,$$

*where $U_1 = L_y^2\eta_t^2 + \frac{L_y\eta_t}{4M_f} + \frac{L_{yx}\eta_t^2}{2\alpha}$, $U_2 = 1 + 4M_f L_y\eta_t + \frac{\alpha L_{yx}\hat{D}_f^2\eta_t^2}{2}$, and $U_3 = \frac{L_y^2\eta_t^2}{m} + \frac{L_{yx}\eta_t^2}{2\alpha m}$ for any $\alpha > 0$.*

*Proof:*

We can get that

$$\mathbb{E}\left[ \left\| y^{t+1} - y^*(x^{t+1}) \right\|^2 \right] = \mathbb{E}\left[ \left\| y^{t+1} - y^*(x^t) \right\|^2 \right] + \mathbb{E}\left[ \left\| y^*(x^{t+1}) - y^*(x^t) \right\|^2 \right] + 2\mathbb{E}\left[ \left\langle y^{t+1} - y^*(x^t), y^*(x^t) - y^*(x^{t+1}) \right\rangle \right]$$

Then,

$$\mathbb{E}\left[ \left\| y^*(x^{t+1}) - y^*(x^t) \right\|^2 \right] \leq L_y^2\mathbb{E}\left[ \left\| x^{t+1} - x^t \right\|^2 \right] \leq L_y^2\mathbb{E}\left[ \left\| \eta_t\overline{H}(x^t, y^{t+1}) \right\|^2 \right] + \frac{\eta_t^2 L_y^2\hat{\sigma}_f^2}{m},$$

and

$$2\mathbb{E}\left[ \left\langle y^{t+1} - y^*(x^t), y^*(x^t) - y^*(x^{t+1}) \right\rangle \right] = -2\mathbb{E}\left[ \left\langle y^{t+1} - y^*(x^t), \nabla y^*(x^t)(x^{t+1} - x^t) \right\rangle \right]$$
$$- 2\mathbb{E}\left[ \left\langle y^{t+1} - y^*(x^t), y^*(x^{t+1}) - y^*(x^t) - \nabla y^*(x^t)(x^{t+1} - x^t) \right\rangle \right]$$

Then, for $-\mathbb{E}\left[ \left\langle y^{t+1} - y^*(x^t), \nabla y^*(x^t)(x^{t+1} - x^t) \right\rangle \right]$, we have

$$-\mathbb{E}\left[ \left\langle y^{t+1} - y^*(x^t), \nabla y^*(x^t)(x^{t+1} - x^t) \right\rangle \right]$$
$$= -\mathbb{E}\left[ \left\langle y^{t+1} - y^*(x^t), \eta_t\nabla y^*(x^t)\overline{H}(x^t, y^{t+1}) \right\rangle \right]$$
$$\leq \mathbb{E}\left[ \left\| y^{t+1} - y^*(x^t) \right\| \left\| \eta_t\nabla y^*(x^t)\overline{H}(x^t, y^{t+1}) \right\| \right]$$

$$\leq L_y \mathbb{E}\left[\left\|y^{t+1} - y^*(x^t)\right\| \left\|\eta_t \overline{H}(x^t, y^{t+1})\right\|\right]$$

$$\leq 2\gamma \mathbb{E}\left[\left\|y^{t+1} - y^*(x^t)\right\|^2\right] + \frac{L_y^2 \eta_t^2}{8\gamma} \mathbb{E}\left[\left\|\overline{H}(x^t, y^{t+1})\right\|^2\right]$$

$$= 2M_f L_y \eta_t \mathbb{E}\left[\left\|y^{t+1} - y^*(x^t)\right\|^2\right] + \frac{L_y \eta_t}{8M_f} \mathbb{E}\left[\left\|\overline{H}(x^t, y^{t+1})\right\|^2\right],$$

where Young's inequality is applied in all inequality and set $\gamma = M_f L_y \eta_t$.

Next, for $-\mathbb{E}\left[\left\langle y^{t+1} - y^*(x^t), y^*(x^{t+1}) - y^*(x^t) - \nabla y^*(x^t)(x^{t+1} - x^t)\right\rangle\right]$, we have

$$-\mathbb{E}\left[\left\langle y^{t+1} - y^*(x^t), y^*(x^{t+1}) - y^*(x^t) - \nabla y^*(x^t)(x^{t+1} - x^t)\right\rangle\right]$$

$$\leq \mathbb{E}\left[\left\|y^{t+1} - y^*(x^t)\right\| \left\|y^*(x^{t+1}) - y^*(x^t) - \nabla y^*(x^t)(x^{t+1} - x^t)\right\|\right]$$

$$\leq \frac{L_{yx}}{2} \mathbb{E}\left[\left\|y^{t+1} - y^*(x^t)\right\| \left\|x^{t+1} - x^t\right\|^2\right]$$

$$\leq \frac{\alpha L_{yx}}{4} \mathbb{E}\left[\left\|y^{t+1} - y^*(x^t)\right\|^2 \left\|x^{t+1} - x^t\right\|^2\right] + \frac{L_{yx}}{4\alpha} \mathbb{E}\left[\left\|x^{t+1} - x^t\right\|^2\right]$$

$$\leq \frac{\alpha L_{yx} \hat{D}_f \eta_t^2}{4} \mathbb{E}\left[\left\|y^{t+1} - y^*(x^t)\right\|^2\right] + \frac{L_{yx} \eta_t^2}{4\alpha} \mathbb{E}\left[\left\|\overline{H}(x^t, y^{t+1})\right\|^2\right] + \frac{L_{yx} \eta_t^2 \hat{\sigma}_f^2}{4m\alpha}$$

Combining the above inequalities and rearranging, we can get that

$$\mathbb{E}\left[\left\|y^{t+1} - y^*(x^{t+1})\right\|^2\right] \leq U_1 \mathbb{E}\left[\left\|\overline{H}(x^t, y^{t+1})\right\|^2\right] + U_2 \mathbb{E}\left[\left\|y^{t+1} - y^*(x^t)\right\|^2\right] + U_3 \hat{\sigma}_f.$$

### A.8 Proof of Theorem 1

From above lemmas, we have

$$\mathbb{E}[W^{t+1}] - \mathbb{E}[W^t] = \Phi(x^{t+1}) - \Phi(x^t) + \frac{M_f}{L_y}\left(\left\|y^{t+1} - y^*(x^{t+1})\right\|^2 - \left\|y^t - y^*(x^t)\right\|^2\right)$$

$$\leq \frac{L_f \eta_t^2}{m}\left(\frac{3}{m}\hat{D}_f \sum_{i=1}^{m} \tau_i^t + 3\hat{\sigma}_f + \frac{12\hat{D}_f}{m}\right) + \eta_t b^2 + \eta_t M_f^2 \mathbb{E}\left[\left\|y^{t+1} - y^*(x^t)\right\|\right]$$

$$\quad + (L_f \eta_t^2 - \frac{\eta_t}{2})\mathbb{E}\left[\left\|\overline{H}(x^t, y^{t+1})\right\|^2\right] - \frac{\eta_t}{2}\mathbb{E}\left[\left\|\nabla\Phi(x^t)\right\|^2\right] + \frac{M_f U_1}{L_y}\mathbb{E}\left[\left\|\overline{H}(x^t, y^{t+1})\right\|^2\right]$$

$$\quad + \frac{M_f U_2}{L_y}\mathbb{E}\left[\left\|y^{t+1} - y^*(x^t)\right\|^2\right] + \frac{M_f U_3 \hat{\sigma}_f^2}{L_y} - \frac{M_f}{L_y}\mathbb{E}\left[\left\|y^t - y^*(x^t)\right\|^2\right]$$

$$= \eta_t b^2 + \left(\frac{\eta_t^2 L_f}{m} + \frac{3M_f U_3}{L_y}\right)\hat{\sigma}_f^2 + \left(\frac{3L_f \eta_t^2}{m^2}\sum_{i=1}^{m}\tau_i^t + \frac{12L_f \eta_t^2}{m^2}\right)\hat{D}_f$$

$$\quad + (L_f \eta_t^2 - \frac{\eta_t}{2} + \frac{M_f U_1}{L_y})\mathbb{E}\left[\left\|\overline{H}(x^t, y^{t+1})\right\|^2\right]$$

$$\quad - \frac{\eta_t}{2}\mathbb{E}\left[\left\|\nabla\Phi(x^t)\right\|^2\right] + \left(\frac{M_f U_2}{L_y} + \eta_t M_f^2\right)\mathbb{E}\left[\left\|y^{t+1} - y^*(x^t)\right\|\right] - \frac{M_f}{L_y}\mathbb{E}\left[\left\|y^t - y^*(x^t)\right\|^2\right]$$

$$\overset{1.1}{\leq} \eta_t b^2 + \left(\frac{\eta_t^2 L_f}{m} + \frac{3M_f U_3}{L_y}\right)\hat{\sigma}_f^2 + \left(\frac{3L_f \eta_t^2}{m^2}\sum_{i=1}^{m}\tau_i^t + \frac{12L_f \eta_t^2}{m^2}\right)\hat{D}_f - \frac{\eta_t}{2}\mathbb{E}\left[\left\|\nabla\Phi(x^t)\right\|^2\right]$$

$$\quad + \left(\frac{M_f U_2}{L_y} + \eta_t M_f^2\right)\mathbb{E}\left[\left\|y^{t+1} - y^*(x^t)\right\|\right] - \frac{M_f}{L_y}\mathbb{E}\left[\left\|y^t - y^*(x^t)\right\|^2\right]$$

$$\overset{1.2}{\leq} \eta_t b^2 + \left(\frac{\eta_t^2 L_f}{m} + \frac{3M_f U_3}{L_y}\right)\hat{\sigma}_f^2 + \left(\frac{3L_f \eta_t^2}{m^2}\sum_{i=1}^{m}\tau_i^t + \frac{12L_f \eta_t^2}{m^2}\right)\hat{D}_f$$

$$\quad - \frac{\eta_t}{2}\mathbb{E}\left[\left\|\nabla\Phi(x^t)\right\|^2\right] - \frac{M_f}{L_y}\mathbb{E}\left[\left\|y^t - y^*(x^t)\right\|^2\right]$$

$$\quad + \left(\frac{M_f U_2}{L_y} + \eta_t M_f^2\right)\left[\left(\prod_{k=0}^{K_m-1}(1 - \beta_{t,k}\mu_g)\right)\mathbb{E}\left[\left\|y^t - y^*(x^t)\right\|^2\right] + \frac{4(\sigma_g^2 + \sigma_{g,1}^2)}{mE^t}\sum_{k=0}^{K_m-1}\beta_{t,k}^2\right]$$

$$=\eta_t b^2 + (\frac{\eta_t^2 L_f}{m} + \frac{3M_f U_3}{L_y})\hat{\sigma}_f^2 + (\frac{3L_f \eta_t^2}{m^2}\sum_{i=1}^{m}\tau_i^t + \frac{12L_f \eta_t^2}{m^2})\hat{D}_f$$

$$-\frac{\eta_t}{2}\mathbb{E}\left[\left\|\nabla\Phi(x^t)\right\|^2\right] + (\frac{M_f U_2}{L_y} + \eta_t M_f^2)\frac{4(\sigma_g^2 + \sigma_{g,1}^2)}{mE^t}\sum_{k=0}^{K_m-1}\beta_{t,k}^2$$

$$+\frac{M_f}{L_y}\left[(U_2 + \eta_t M_f L_y)\left(\prod_{k=0}^{K_m-1}(1 - \beta_{t,k}\mu_g)\right) - 1\right]\mathbb{E}\left[\left\|y^t - y^*(x^t)\right\|^2\right]$$

$$(5)$$

where 1.1 follows the fact that $L_f\eta_t^2 - \frac{\eta_t}{2} + \frac{M_f U_1}{L_y} \leq 0$ if $\eta_t \leq (2L_f + 4M_f L_y + \frac{2M_f L_{yx}}{L_y\alpha})^{-1}$, 1.2 uses Lemma 2.

If we choose $\beta_{t,k} \leq \frac{1}{\mu_g}$ and $K_m$ be an oven number, then equation 5 can be transferred to

$$\mathbb{E}[W^{t+1}] - \mathbb{E}[W^t] \leq \eta_t b^2 + (\frac{\eta_t^2 L_f}{m} + \frac{3M_f U_3}{L_y})\hat{\sigma}_f^2 + (\frac{3L_f \eta_t^2}{m^2}\sum_{i=1}^{m}\tau_i^t + \frac{12L_f \eta_t^2}{m^2})\hat{D}_f$$

$$-\frac{\eta_t}{2}\mathbb{E}\left[\left\|\nabla\Phi(x^t)\right\|^2\right] + (\frac{M_f U_2}{L_y} + \eta_t M_f^2)\frac{4(\sigma_g^2 + \sigma_{g,1}^2)}{mE^t}\sum_{k=0}^{K_m-1}\beta_{t,k}^2$$

Then telescoping from $t = 0$ to $t = T - 1$, we can get

$$\frac{1}{T}\left[\mathbb{E}[W^t] - \mathbb{E}[W^0]\right] \leq b^2 + \frac{1}{T}\sum_{t=0}^{T-1}(\frac{\eta_t^2 L_f}{m} + \frac{3M_f U_3}{L_y})\hat{\sigma}_f^2 + \frac{1}{T}\sum_{t=0}^{T-1}(\frac{3L_f \eta_t^2}{m^2}\sum_{i=1}^{m}\tau_i^t + \frac{12L_f \eta_t^2}{m^2})\hat{D}_f$$

$$-\frac{1}{T}\sum_{t=0}^{T-1}\frac{\eta_t}{2}\mathbb{E}\left[\left\|\nabla\Phi(x^t)\right\|^2\right] + \frac{1}{T}\sum_{t=0}^{T-1}(\frac{M_f U_2}{L_y} + \eta_t M_f^2)\frac{4(\sigma_g^2 + \sigma_{g,1}^2)}{mE^t}\sum_{k=0}^{K_m-1}\beta_{t,k}^2$$

Then, if let $\bar{\eta} = \frac{1}{T}\left(\sum_{t=0}^{T-1}\eta_t\right)$, and using Chebyshev's Inequality, we have $\frac{1}{T}\left(\sum_{t=0}^{T-1}\eta_t\right)\left(\sum_{t=0}^{T-1}\mathbb{E}\left[\left\|\nabla f(x^t)\right\|^2\right]\right) \leq \sum_{t=0}^{T-1}\eta_t\mathbb{E}\left[\left\|\nabla f(x^t)\right\|^2\right]$. Thus, we can rewrite above equation as

$$\frac{1}{T}\sum_{t=0}^{T-1}\mathbb{E}\left[\left\|\nabla\Phi(x^t)\right\|^2\right] \leq \frac{2}{\bar{\eta}T}\left[\mathbb{E}[W^0] - \mathbb{E}[W^*]\right] + \frac{2}{\bar{\eta}T}\sum_{t=0}^{T-1}(\frac{M_f U_2}{L_y} + \eta_t M_f^2)\frac{4(\sigma_g^2 + \sigma_{g,1}^2)}{mE^t}\sum_{k=0}^{K_m-1}\beta_{t,k}^2$$

$$+ b^2 + \frac{1}{\bar{\eta}T}\sum_{t=0}^{T-1}(\frac{\eta_t^2 L_f}{m} + \frac{3M_f U_3}{L_y})\hat{\sigma}_f^2 + \frac{1}{\bar{\eta}T}\sum_{t=0}^{T-1}(\frac{3L_f \eta_t^2}{m^2}\sum_{i=1}^{m}\tau_i^t + \frac{12L_f \eta_t^2}{m^2})\hat{D}_f$$

## B EXPERIMENTS

All experiments are implemented in Matlab 2023a onana ASUS laptop with an Nvidia GeForce GTX GPU. Note that the current experiments and the results in other related works are all simulations on a single laptop and simulate for distributed communication. The linear speedup improvement can be shown by implementing the model and the algorithms on a distributed setting with multiple machines.

To prove the efficiency of the proposed AFBO algorithm, we use a similar setting in (Jiao et al., 2023).

### B.1 SIMULATION SETUP

Experiments are conducted on hyper-parameter optimization tasks (i.e., data hyper-cleaning task and regularization coefficient optimization task) in the distributed setting to evaluate the performance of

the proposed AFBO. The proposed AFBO is compared with the state-of-the-art distributed bilevel optimization method FedNest (Tarzanagh et al., 2022) and ADBO (Jiao et al., 2023). In the data hyper-cleaning task, experiments are carried out on MNIST (LeCun et al., 1998). In coefficient optimization task, following (Chen et al., 2022a), experiments are conducted on Covertype (Blackard & Dean, 1999).

In the MNIST dataset to impose data heterogeneity, we split the data based on the classes ($p$) of images each client contains (McMahan et al., 2017; Li et al., 2020a; Yang et al., 2021a). We distribute the data to $M$ clients such that each client contains only certain $p$ classes with the same number of training/test samples. Specifically, each client randomly chooses $p$ classes of labels and evenly samples training/testing data points only with these $p$ class labels from the whole dataset without replacement. For example, for $p = 2$, each client only has training/testing samples with 2 classes, which causes heterogeneity among different clients. For $p = 10$, each client has samples with 10 classes, which is nearly the IID case. In this way, we can use $p$ to represent the non-IID degree qualitatively (i.e., the smaller $p$ is the higher the non-IID degree is).

## B.2 SIMULATION RESULTS

### B.2.1 DATA HYPER-CLEANING TASK

Following (Ji et al., 2021; Yang et al., 2021b), the proposed AFBO algorithm is compared with ADBO (Jiao et al., 2023) and distributed bilevel optimization method FedNest (Tarzanagh et al., 2022) on the distributed data hyper-cleaning task (Chen et al., 2022b) on MNIST datasets (LeCun et al., 1998). Data hyper-cleaning (Chen et al., 2022b) involves training a classifier in a contaminated environment where each training data label is changed to a random class number with a probability (i.e., the corruption rate). In addition, we further consider the effect of heterogeneous data distribution on the training performance. The distributed data hyper-cleaning problem can be expressed as,

$$\min F(\boldsymbol{\phi}, \boldsymbol{\omega}) = \sum_{i=1}^{m} \frac{1}{|D_i^{val}|} \sum_{(\boldsymbol{x}_j, y_j) \in D_i^{val}} L(\boldsymbol{x}_j^T \boldsymbol{\omega}, y_j)$$

$$s.t. \ \boldsymbol{\omega} = \arg\min_{\boldsymbol{\omega}'} f(\boldsymbol{\phi}, \boldsymbol{\omega}') = \sum_{i=1}^{m} \frac{1}{|D_i^{tr}|} \sum_{(\boldsymbol{x}_j, y_j) \in D_i^{tr}} \sigma(\phi_j) L(\boldsymbol{x}_j^T \boldsymbol{\omega}', y_j) + \|\boldsymbol{\omega}'\|^2$$

where $D_i^{tr}$ and $D_i^{val}$ denote the training and validation datasets on $i$-th client, espectively. $(\boldsymbol{x}_j, y_j)$ denotes the $j$-th data and label. $\sigma(.)$ is the sigmoid function, $L$ is the cross-entropy loss, and $m$ is the number of clients in the distributed system. In the MNIST dataset, we set $m = 18$ and $\tau = 10$. As in (Cohen et al., 2021), we assume that the communication delay of each client obeys the heavy-tailed distribution. The proposed AFBO is compared with the state-of-the-art distributed bilevel optimization method FedNest, ADBO and SDBO (Synchronous Distributed Bilevel Optimization, i.e., ADBO without asynchronous setting). The test accuracy results of the 4 algorithms with IID and non-IID datasets are shown in Fig. 1 and Fig. 2. We can observe that the proposed AFBO is the most efficient algorithm. Since the asynchronous setting is considered in AFBO, the server can update its variables once it receives updates from updating clients, It allows multiple local iterations, which makes full use of clients' computing resources.

### B.2.2 REGULARIZATION COEFFICIENT OPTIMIZATION TASK

Following (Chen et al., 2022a), we compare the performance of AFBO with baseline algorithms FedNest, SDBO, and ADBO on the regularization coefficient optimization task using Covertype datasets. The distributed regularization coefficient optimization problem is defined as,

$$\min F(\boldsymbol{\phi}, \boldsymbol{\omega}) = \sum_{i=1}^{m} \frac{1}{|D_i^{val}|} \sum_{(\boldsymbol{x}_j, y_j) \in D_i^{val}} L(\boldsymbol{x}_j^T \boldsymbol{\omega}, y_j)$$

$$s.t. \ \boldsymbol{\omega} = \arg\min_{\boldsymbol{\omega}'} f(\boldsymbol{\phi}, \boldsymbol{\omega}') = \sum_{i=1}^{m} \frac{1}{|D_i^{tr}|} \sum_{(\boldsymbol{x}_j, y_j) \in D_i^{tr}} L(\boldsymbol{x}_j^T \boldsymbol{\omega}', y_j) + \sum_{j=1}^{n} \phi_j(\omega_j')$$

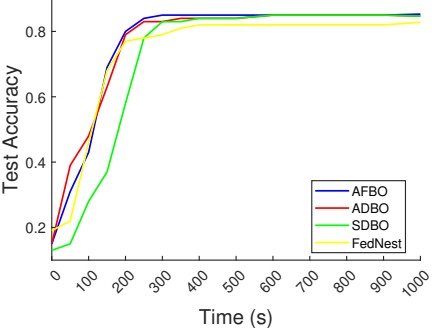

Figure 5: Test accuracy vs time on IID MINIST.

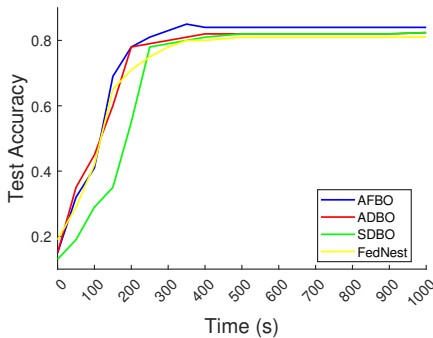

Figure 6: Test accuracy vs time on non-IID MINIST.

where $\phi \in \mathbb{R}^n$, $\omega \in \mathbb{R}^n$ and $L$, respectively denote the regularization coefficient, model parameter, and logistic loss, and $\omega' = [\omega'_1, ..., \omega'_n]$. In the regularization coefficient optimization task with the Covertype dataset, we set $N = 18$ and $\tau = 15$. We also assume that the delay of each client obeys the heavy-tailed distribution. Firstly, we compare the performance of the proposed AFBO, ADBO, SDBO, and FedNest in terms of test accuracy on the Covertype dataset. The results on the Covertype dataset are shown in Fig. 3, which shows that AFBO achieves the best performance among all the schemes.

Next, we assume there are at most five stragglers in the distributed system, and the mean of (communication + computation) delay of stragglers is five times the delay of normal clients. The result is shown in Fig. 4. It is found that the efficiency of the synchronous distributed algorithms (FedNest and SDBO) has been significantly affected, while the proposed AFBO and ADBO suffer slightly from the straggler problem since they are asynchronous methods and only consider the updating clients.

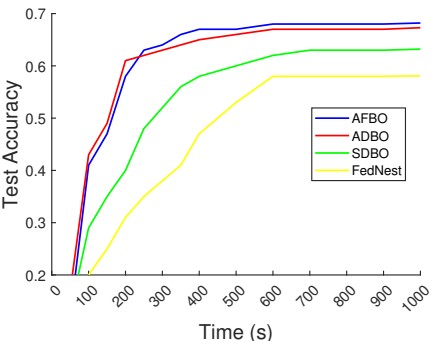

Figure 7: Test accuracy vs. time on Covertype.

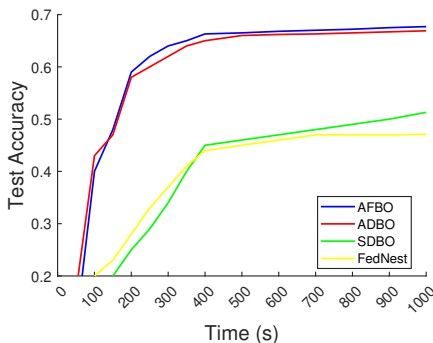

Figure 8: Test accuracy vs. time on Covertype.

### B.3 CONVERGENCE OF DIFFERENT ALGORITHMS UNDER DIFFERENT INNER AND OUTER LOOP DELAYS

In the previous sections, we assume there are only outer loop asynchronous delays, it shows that AFBO performs similarly to the ADBO and AFBO performs better than other algorithms. In this section, we allow there exists both inner loop and outer loop asynchronous delays. In the MNIST dataset, we set $m = 18$, $\tau = 10$, and $\rho = 10$. In the regularization coefficient optimization task with the Covertype dataset, we set $N = 18$, $\tau = 15$, and $\rho = 10$. As in (Cohen et al., 2021), we assume

that the communication delay of each client obeys the heavy-tailed distribution. We first compare AFBO, ADBO, SDBO, FedNest, and Prometheus (Liu et al., 2023) algorithms' performance on the IID MINIST dataset, then we conduct the experiments on the non-IID MINIST dataset and the Covertype dataset.

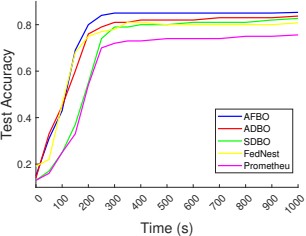 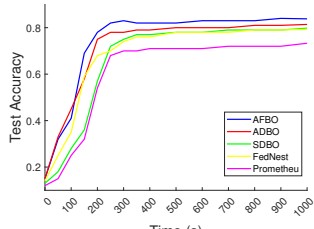 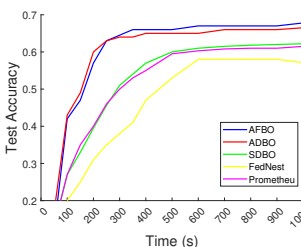

Figure 9: Test accuracy vs time on IID MINIST.

Figure 10: Test accuracy vs time on non-IID MINIST.

Figure 11: Test accuracy vs time on Covertype.

Fig. 9, fig. 10, and Fig. 11 show that AFBO performs best among all algorithms. In addition, it shows that all algorithms expect AFBO to suffer an obvious convergence degeneration as there exists an inner loop asynchronous delay. Among them, synchronous algorithms (e.g., SDBO, FedNest, and Prometheus) degenerate more than AFBO and ADBO. Moreover, from Fig. 9 and fig. 10, we can find that convergence degeneration is much more obvious in the non-IID MINIST dataset than in the IID MINIST dataset. This is because the bias of using the most recent update of a client is much larger in the non-IID MINIST dataset case than in the IID MINIST dataset case. Intuitively, a reusing of past gradients inducts a new bias of SGD, as some of the SGD gradients use more than others. Finally, Fig. 9 and Fig. 11, show that the convergence degeneration in the Covertype dataset is slighter than in the MINIST dataset.

### B.4 ROUND CONVERGENCE OF DIFFERENT ALGORITHMS

In this subsection, we compare the convergence rate vs round of AFBO, ADBO, SDBO, FedNest, and Prometheus (Liu et al., 2023). We allow there exists both inner loop and outer loop asynchronous delays. In the MNIST dataset, we set $m = 18$, $\tau = 10$, and $\rho = 10$. In the regularization coefficient optimization task with the Covertype dataset, we set $N = 18$, $\tau = 15$, and $\rho = 10$. As in (Cohen et al., 2021), we assume that the communication delay of each client obeys the heavy-tailed distribution. For AFBO, ADBO, SDBO, and FedNest algorithm, one global round means the server updates the global model. For the Prometheus algorithm, one global round means the server communicates with clients.

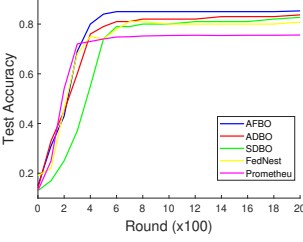 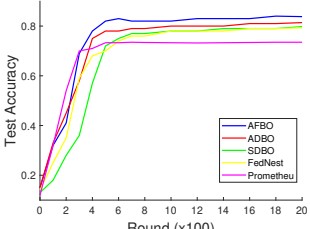 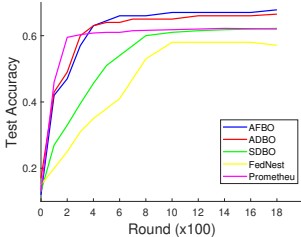

Figure 12: Test accuracy vs rounds on IID MINIST.

Figure 13: Test accuracy vs rounds on non-IID MINIST.

Figure 14: Test accuracy vs rounds on Covertype.

Fig. 12, Fig. 13, and Fig. 14 show that the proposed AFBO algorithm performs best among all baseline algorithms. It can be found that there is nearly no difference in convergence rates of AFBO, ADBO, SDBO, and FedNest. However, there are some differences in the Prometheus algorithm. This is because the Prometheus algorithm communicates with the server fewer than the rest algo-

rithms. Based on the definition of one global round in this subsection, the figure of the Prometheus algorithm is different from Fig. 9, fig. 10, and Fig. 11. Although the figures have some differences, the final test accuracy of these algorithms is similar to Fig. 9, fig. 10, and Fig. 11. These results show that though AFBO provides the most flexibility to clients, the convergence rate and accuracy are stable.

## B.5 COMPARISON WITH AFL

AFL (Yang et al., 2022b) is a well-known algorithm which allows clients flexible participation. Our proposed AFBO also allows clients to have the maximum flexibility to participate in the FBO training. In this subsection, we compare the performance of AFL and AFBO. We use AFL and AFBO to perform the data hyper-cleaning task in IID and non-IID MINIST datasets. We set $m = 18$, $\tau = 10$, and $\rho = 10$ for AFBO algorithm and set $m = 18$ and $\tau = 10$ for AFL algorithm.

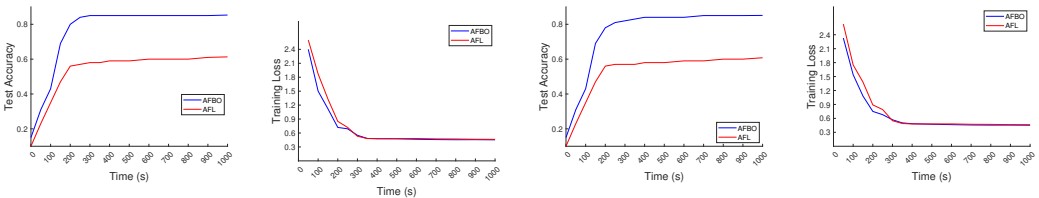

Figure 15: Test accuracy vs time on IID MINIST.

Figure 16: Training loss vs time on IID MINIST.

Figure 17: Test accuracy vs time on non-IID MINIST.

Figure 18: Training loss vs time on non-IID MINIST.

From Fig 15, Fig 16, Fig 17, and Fig 18, we can get that AFL performs worse in FBO tasks. This is because AFL does not design a distributed estimator for the hyper-parameter ($\nabla\Phi$). The difference between $\frac{1}{M}\sum_{i=1}^{M}\frac{\partial F_i(x,y)}{\partial y}\frac{\partial y}{\partial x}$ and $\frac{\partial F(x,y)}{\partial y}\frac{\partial y}{\partial x}$ leads to the low training accuracy of AFL performing FBO tasks. From the training loss aspect, it shows that the convergence rates and speeds of AFBO and AFL are similar. The reason is that both AFL and AFBO use the most recent gradients stored in the server memory, and both algorithms can achieve linear speed-up. Moreover, both AFL and AFBO consider the effect of the dataset's non-IID degree, thus Fig 15 and Fig 17 show that there is little convergence degeneration as the dataset's non-IID degree increases.

## B.6 IMPACT OF ASYNCHRONOUS DELAY

In this subsection, we evaluate the impact of asynchronous delay on the AFBO algorithm. We first conduct the data hyper-cleaning task on the IID and non-IID MINIST datasets, and then we perform the regularization coefficient optimization task on the Covertype dataset. In experiments, we set the total number of clients as 18, and we set the $\tau = 0$, 1, 5, 10, respectively. The impact of the asynchronous delays on the AFBO algorithm is shown in Fig. 19, Fig. 20, and Fig. 21.

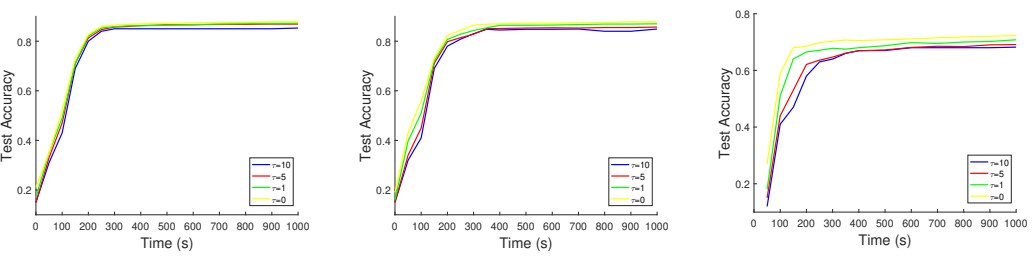

Figure 19: Test accuracy vs time on IID MINIST.

Figure 20: Test accuracy vs time on non-IID MINIST.

Figure 21: Test accuracy vs time on Covertype.

It shows that the test accuracy becomes smaller as the asynchronous delay increases, which matches our theoretical results. In addition, there is little convergence degradation due to the asynchronous delay, which proves the effectiveness and robustness of AFBO. Moreover, Fig. 19 and Fig. 20 show that the convergence degradation under the non-IID MINIST dataset is slightly higher than under the IID MINIST dataset. This is because the contribution of all clients is equal in the IID dataset which means the straggler has low effects on the test accuracy, but some clients owning unique datasets under non-IID dataset settings with higher asynchronous delay can cause a larger convergence degradation due to straggler effects.