# OpenReview forum: "Anarchic Federated Bilevel Optimization"
_ICLR.cc/2024/Conference — Submitted to ICLR 2024_

### Official Review · Reviewer_3Fd8 · 2023-10-30

**Soundness:** 2 fair
**Presentation:** 3 good
**Contribution:** 3 good
**Rating:** 8
**Confidence:** 3

**Summary:**

A double-loop scheme Anarchic Federated Bilevel Optimization (AFBO) is proposed in this work, which allows clients to flexibly participate in federated bilevel optimization training according to their heterogeneous and time-varying computation and communication capabilities. Moreover, theoretical analysis is conducted to show the convergence rate of the proposed method, i.e., it is demonstrated that the proposed  AFBO algorithm can achieve a convergence rate of $O(\sqrt{1/T})$.

**Strengths:**

1. This proposed method is efficient since the clients in the distributed system can participate in any inner or outer rounds, asynchronously, and with any number of local iterations.

2. The authors conduct lots of theoretical analysis about the proposed method, e.g., convergence analysis. It is demonstrated that the proposed  AFBO algorithm can achieve a convergence rate of $O(\sqrt{1/T})$.

3. This paper is well-organized and easy to follow.

**Weaknesses:**

I have some concerns about the experiments and communication complexity as follows.

1. In the experiments, the authors claim the proposed AFBO is the most efficient algorithm. However, more explanation should be added about why the proposed method is more efficient than ADBO in the experiment since the ADBO is also an asynchronous algorithm.

2. More experimental results should be added to show the excellent performance of the proposed AFBO.

3. Lack of analysis for the communication complexity of the proposed method.

**Questions:**

My questions are about the experiments and communication complexity, please see the weakness above.

---

> ### Author Response · Authors · 2023-11-21
>
> Thank you for your comments.
>
> W1: In the experiments, the authors claim the proposed AFBO is the most efficient algorithm. However, more explanation should be added about why the proposed method is more efficient than ADBO in the experiment since the ADBO is also an asynchronous algorithm.
>
> A1: Based on your comment, in the revised paper, we have added more experiment results in Section B.3, Section B.4, Section B.5, and Section B.6. of the supplement material. We conduct experiments on the convergence of different algorithms under different inner and outer loop delays, comparison of AFBO and AFL, and the impact of asynchronous delays, etc., discussed as follows. For the inner loop asynchronous delays experiments, the results show that AFBO performs best among all algorithms. In addition, it shows that all algorithms expect AFBO to suffer an obvious convergence degeneration as there exists an inner loop asynchronous delay. Among them, synchronous algorithms (e.g., SDBO, FedNest, and Prometheus) degenerate much more than AFBO and ADBO. Moreover, we find that convergence degeneration is much more obvious in the non-IID MINIST dataset than in the IID MINIST dataset. This is because the bias of using the most recent update of a client is much larger in the non-IID MINIST dataset case than in the IID MINIST dataset case. Intuitively, a reusing of past gradients inducts a new bias of SGD, as some of the SGD gradients use more than others. Finally, it shows that the convergence degeneration in the Covertype dataset is slighter than in the MINIST dataset. The experiments on comparison between AFBO and AFL shows that AFL performs worse on bilevel tasks than AFBO, but they have a similar convergence rate as they both use the most recent updates to update the global model and achieve a linear speed-up. The experiment on the impact of asynchronous delays on AFBO shows that the test accuracy becomes smaller as the asynchronous delay increases, which matches our theoretical results. In addition, there is little convergence degradation due to the asynchronous delay, which proves the effectiveness and robustness of AFBO. Moreover, it shows that the convergence degradation under the non-IID MINIST dataset is slightly higher than under the IID MINIST dataset. This is because the contribution of all clients is equal in the IID dataset which means the straggler has low effects on the test accuracy, but some clients owning unique datasets under non-IID dataset settings with higher asynchronous delay can cause a larger convergence degradation due to straggler effects.
>
> W2: More experimental results should be added to show the excellent performance of the proposed AFBO.
>
> A2: Based on your comment, in the revised paper, we have added more experiment results in Section B.3, Section B.4, Section B.5, and Section B.6 of the supplement material. Please refer to our response to A1 for detailed discussion of these results.
>
> W3: Lack of analysis for the communication complexity of the proposed method.
>
> A3: Based on your comment, in the revised paper, we have added analysis of the communication complexity of the AFBO algorithm (in Corollary $1$'s Remark, marked in blue): “In addition, we need $T=O(\kappa^5_g\epsilon^{-2}/n)$ to achieve an $\epsilon$-accurate stationary point. Compared with FedNest (Tarzanagh et al., 2022), our complexity has the same dependence on $\kappa$ and $\epsilon$, but a better dependence on $n$ due to the linear speedup."

---

> > ### Comment · Reviewer_3Fd8 · 2023-11-22
> >
> > Thanks for your responses. My concerns have been addressed.

---

> > > ### Author Response · Authors · 2023-11-22
> > >
> > > Thank you.

---

### Official Review · Reviewer_3dLq · 2023-11-01

**Soundness:** 3 good
**Presentation:** 3 good
**Contribution:** 3 good
**Rating:** 5
**Confidence:** 5

**Summary:**

This paper presents a new algorithm called Asynchronous Federated Bilevel Optimization (AFBO), offering a flexible approach for client participation in federated bilevel optimization (FBO) training. The unique aspects of AFBO include the ability for clients to not only join in at any stage of the inner or outer optimization rounds, but also undertake a variable number of local training iterations. The training process can be engaged asynchronously. Rigorous theoretical examination has been conducted to reveal convergence rate. It is seen that AFBO's convergence rate aligning with other benchmarks.

**Strengths:**

Asynchronous federated learning is an important problem with numerous applications. The theoretic analysis is solid and authors have conducted experiments to assess the performance of the proposed algorithm.

**Weaknesses:**

The experiment results are limited. Authors are recommended to compare the performance of the proposed algorithm with algorithms such as [1].  In addition, the idea itself is similar to [2]. Authors are therefore recommended to elaborate more on the difference between this work and existing ones in the literature. In addition, in all figures, it is seen that the performance of AFBO is only slighted better than ADBO.

[1] Prometheus: Taming sample and communication complexities in constrained decentralized stochastic bilevel learning. ICML 2023
[2] Anarchic Federated Learning ICML 2022

**Questions:**

I understand that AFBO offers flexibility by allows clients to engage in the updating in an asynchronous manner.  Is it possible that under such a setting, the AFBO algorithm may converge to a solution that is different from other algorithms ?

---

> ### Author Response · Authors · 2023-11-21
>
> Thank you for your comments.
>
> W1: The experiment results are limited. Authors are recommended to compare the performance of the proposed algorithm with algorithms such as [1]. In addition, the idea itself is similar to [2]. Authors are therefore recommended to elaborate more on the difference between this work and existing ones in the literature. In addition, in all figures, it is seen that the performance of AFBO is only slightly better than ADBO.
>
> [1] Prometheus: Taming sample and communication complexities in constrained decentralized stochastic bilevel learning. ICML 2023  [2] Anarchic Federated Learning ICML 2022
>
> A1:  Based on your comment, in the revised paper, we have added more experiment results in Section B.3, Section B.4, Section B.5, and Section B.6. of the supplement material, discussed as follows. We conduct experiments on the convergence of different algorithms under different inner and outer loop delays, for comparison of AFBO and AFL, and for the impacts of asynchronous delays, etc. For the inner loop asynchronous delays experiments, the results show that AFBO performs best among all algorithms. In addition, it shows that all algorithms expect AFBO to suffer an obvious convergence degeneration as there exists an inner loop asynchronous delay. Among them, synchronous algorithms (e.g., SDBO, FedNest, and Prometheus) degenerate much more than AFBO and ADBO. Moreover, we find that convergence degeneration is much more obvious in the non-IID MINIST dataset than in the IID MINIST dataset. This is because the bias of using the most recent update of a client is much larger in the non-IID MINIST dataset case than in the IID MINIST dataset case. Intuitively, a reusing of past gradients inducts a new bias of SGD, as some of the SGD gradients use more than others. Finally, it shows that the convergence degeneration in the Covertype dataset is slighter than in the MINIST dataset. The experiments on comparison between AFBO and AFL shows that AFL performs worse on bilevel tasks than AFBO, but they have a similar convergence rate as they both use the most recent updates to update the global model and achieve a linear speed-up. The experiment on the impact of asynchronous delays on AFBO shows that the test accuracy becomes smaller as the asynchronous delay increases, which matches our theoretical results. In addition, there is little convergence degradation due to the asynchronous delay, which proves the effectiveness and robustness of AFBO. Moreover, it shows that the convergence degradation under the non-IID MINIST dataset is slightly higher than under the IID MINIST dataset. This is because the contributions of all clients are equal in the IID dataset setting which means the straggler has low effects on the test accuracy, but some clients owning unique datasets under non-IID dataset settings with higher asynchronous delay can cause a larger convergence degradation due to straggler effects.
>
> There are some major differences of this paper compared to [2], discussed as follows (we have added some discussion on the differences compard to [2] in Section 2.1 of the revised paper, marked in blue). While [2] considered conventional single-level learning, the AFBO algorithm of this paper focuses on a bilevel learning problem, which is very different and much more complicated than the single-level learning problem. Specifically, bilevel optimization needs to estimate the hyper-gradient $\nabla \Phi$. In the distributed setting, this is more challenging than for the single-level learning, since we have $\frac1M\sum_{i=1}^M\frac{\partial F_i(x,y)}{\partial y}\frac{\partial y}{\partial x}\neq\frac{\partial F(x,y)}{\partial y}\frac{\partial y}{\partial x}$, so that $\nabla \Phi \neq \frac1n\sum_{i=1}^N \nabla \Phi_i$. As a result, AFBO needs to estimate $\nabla \Phi$ using a distributed estimator, given by $H_i(x^t,y^{t+1})=\triangledown_x f_i(x^t,y^{t+1};\phi_i^t)-\triangledown^2_{xy}g_i(x^t,y^{t+1};\rho_i^t)\times\left[\frac{N}{l_{g,1}}\prod_{l=1}^{N_i}(I-\frac1{l_{g,1}}\triangledown_{yy}^2g_i(x^t,y^{t+1};\zeta_i^t))\right]\times\triangledown_yf_i(x^t,y^{t+1};\xi_i^t)$. Then we can aggregate $H_i(x^t,y^{t+1})$ to estimate $\nabla \Phi$. Therefore, the AFBO algorithm involves two loops: an inner loop that updates the lower-level variable $y^{t,k}$, and an outer loop that updates the upper-level variable $x^t$. This is a very different algorithm structure compared to the AFL algorithm in [2] which only involves one loop. Moreover, the AFBO algorithm allows for maximum freedom for clients by taking into account various anarchic settings: partial participation and asynchronous participation in both the inner and outer loops, and also different local iteration numbers in the inner loop.

---

> > ### Author Response · Authors · 2023-11-21
> >
> > Therefore, the convergence analysis of AFBO also has non-trivial differences compared to that in [2]: we need to first characterize the convergence error under those anarchic factors for the inner loop, and then characterize the convergence error under the anarchic factors for the outer loop, which depends on that for the inner loop. Furthermore, the convergence results of AFBO (in Theorem 1 and Corollary 1) show the impacts of various anarchic settings for both the inner and outer loops of the algorithm, which are very different from those in [2].
> >
> > For the performance of AFBO, our algorithm focuses on flexibility and communication efficiency, where we allow that there exists both an inner loop and an outer loop asynchronous delay, while ADBO does not allow both delays. We provide a comparison of different algorithms in the supplement material Section B.3. The experiment results show that AFBO performs best among all algorithms. In addition, it shows that all algorithms expect AFBO to suffer an obvious convergence degeneration as there exists an inner loop asynchronous delay. Among them, synchronous algorithms (e.g., SDBO, FedNest, and Prometheus) degenerate much more than AFBO and ADBO. Moreover, we find that convergence degeneration is much more obvious in the non-IID MINIST dataset than in the IID MINIST dataset. This is because the bias of using the most recent update of a client is much larger in the non-IID MINIST dataset case than in the IID MINIST dataset case. Intuitively, reusing past gradients inducts a new bias of SGD, as some of the SGD gradients use more than others.
> >
> > Q1: I understand that AFBO offers flexibility by allowing clients to engage in the updating in an asynchronous manner. Is it possible that under such a setting, the AFBO algorithm may converge to a solution that is different from other algorithms?
> >
> > A2: This paper focuses on non-convex-strongly-convex bilevel problems. Due to the non-convex upper-level objective function, it is standard for existing works in this setting (such as [3]-[7]) that the result of the convergence analysis can only guarantee that the average gradient norm over all rounds can be made arbitrarily small. This means that the algorithm does not guarantee to converge to any particular solution of the learning problem.
> >
> > For our future work, we can study strongly-convex-strongly-convex bilevel problems. In this case, we may be able to show that the AFBO algorithm can converge to the optimal solution of the learning problem. As the optimal solution is unique for a strongly convex problem, all algorithms that can converge to the optimal solution converge to the same solution.
> >
> > [3] "Achieving Linear Speedup in Non-IID Federated Bilevel Learning", ICML'23
> >
> > [4] "SimFBO: Towards Simple, Flexible and Communication-efficient Federated Bilevel Learning", NeurIPS'23.
> >
> > [5] "Towards Gradient-based Bilevel Optimization with Non-convex Followers and Beyond", NeurIPS' 21.
> >
> > [6] "Communication-Efficient Federated Bilevel Optimization with Global and Local Lower Level Problems", NeurIPS' 23.
> >
> > [7] "Communication-Efficient Federated Hypergradient Computation via Aggregated Iterative Differentiation", ICML' 23.

---

> ### Author Response · Authors · 2023-11-22
>
> Dear reviewer: We have added responses to your comments. Could you take a look and let us know if you have any further comment? Thanks.

---

> ### Author Response · Authors · 2023-11-23
>
> Dear reviewer: We have added responses to your comments. Could you take a look and let us know if you have any further comment? Thanks.

---

> ### Comment · Area_Chair_rNtp · 2023-12-04
>
> Dear Reviewer 3dLq,
>
> The authors have answered your comments. Please read their responses as soon as possible (if you haven't) and reply here if you think the authors have addressed your concerns (maybe some of them). For example, I personally think the setting of  [2]  is definitely different from this paper and the challenge the authors mention make sense. Of course, I am not sure if the technical analysis (provig technique) is significantly different due to the new challenge, which I expect you to judge as a reviewer. Please update your rating if you have changed your evaluation, or mention that you want to keep your rating.
>
> Thank you so much.
>
>
>
> Area Chair

---

### Official Review · Reviewer_PRRM · 2023-11-02

**Soundness:** 1 poor
**Presentation:** 1 poor
**Contribution:** 1 poor
**Rating:** 3
**Confidence:** 5

**Summary:**

This paper studied federated bilevel optimization under the asynchronous setting. However, there are some fatal errors in convergence analysis.

**Strengths:**

The problem investigated is important.

The writing is good.

**Weaknesses:**

1. This paper used impractical assumptions. In particular, it assumes the gradient of $g$ is upper bounded and $g$ is strongly convex.  A strongly convex quadratic function does not satisfy those two assumptions simultaneously.

2. There are some fatal errors. In particular, this paper denotes
$\bar{H}\left(x^t, y^{t+1}\right):=\mathbb{E}\left[\frac{1}{m} \sum_{i \in \mathcal{M}} \bar{H}_i\left(x^{t-\tau_i^t}, y^{t-\tau_i^t+1}\right)\right]$. Then, in convergence analysis, the authors directly use $x^{t}$ rather than $x^{t-\tau_i^t}$, e.g., the second equation when proving lemma 1. THis is totally wrong.

3. For an asynchronous algorithm, how the communication latency affects the convergence rate should be discussed.

**Questions:**

1. This paper used impractical assumptions. In particular, it assumes the gradient of $g$ is upper bounded and $g$ is strongly convex.  A strongly convex quadratic function does not satisfy those two assumptions simultaneously.

2. There are some fatal errors. In particular, this paper denotes
$\bar{H}\left(x^t, y^{t+1}\right):=\mathbb{E}\left[\frac{1}{m} \sum_{i \in \mathcal{M}} \bar{H}_i\left(x^{t-\tau_i^t}, y^{t-\tau_i^t+1}\right)\right]$. Then, in convergence analysis, the authors directly use $x^{t}$ rather than $x^{t-\tau_i^t}$, e.g., the second equation when proving lemma 1. THis is totally wrong.

3. For an asynchronous algorithm, how the communication latency affects the convergence rate should be discussed.

---

> ### Author Response · Authors · 2023-11-21
>
> Thank you for your comments!
>
> Q1: This paper used impractical assumptions. In particular, it assumes the gradient of g is upper bounded and g is strongly convex. A strongly convex quadratic function does not satisfy those two assumptions simultaneously.
>
> A1: First, we do not assume that the gradient of the lower-level local objective function $g_i$ is upper bounded. We only assume (in Assumption 4) that the variances of the 1st order and 2nd order stochastic gradients of function $g_i$ is bounded (i.e., $E_{\zeta}\left[ \left\|\triangledown g_i(z; \xi) - \triangledown g_i(z)\right\|^2\right]\le  \sigma^2_{g,1}$, $E_{\zeta}\left[ \left\|\triangledown^2 g_i(z; \xi) - \triangledown^2 g_i(z)\right\|^2\right]\le  \sigma^2_{g,2}$). This is a common assumption used in existing works on bilevel optimization (BO) including federated bilevel optimization (FBO), such as [1]-[5], [7]-[10]. For clarity, we have renamed Assumption 4 as “Bounded variance of local stochastic gradient”.
>
> Another assumption (Assumption 5) we make for function $g_i$ is that the variance between local gradients and the global gradient is bounded (i.e., $\mathbb{E} \left [\left\|\triangledown g_i(z) - \triangledown g(z)\right\|^2 \right] \le \sigma_g^2$). This is also a widely used assumption in existing works on BO including FBO, such as [1]-[4], [7]-[9].
>
> In addition, in Lemma 6 we show that the second-order moment of the hyper-gradient $H_i$ is upper bounded, i.e., $\mathbb{E}\left[\left\|H_i(x^t,y^{t+1}) \right\|^2|F^t\right] \le \hat{D}_f$. Note that this is not an assumption, but it is a result derived from Assumption 4. This is a widely used result in existing works on BO including FBO, such as [2]-[5], [7]-[10].
>
> Indeed, you are correct that for an arbitrary strongly convex function $g$ such as a quadratic function, its gradient norm cannot be upper bounded. However, our paper does not assume that the gradient norm of the low-level objective function $g$ is bounded, and thus it does not contradict the assumption that $g$ is strongly convex.
>
> Moreover, a few recent works, such as [6], consider a non-strongly-convex lower-level objective function $g$ (which can be non-convex or weakly-convex). In this setting, it is challenging to find the optimal solution of the lower-level optimization problem (there may exist multiple optimal solutions). Therefore, the convergence analysis in these works usually cannot achieve a reasonable performance guarantee. We will explore AFBO for the non-strongly-convex setting in our future work.
>
> [1] "Anarchic Federated Learning", ICML'22
>
> [2] "Achieving Linear Speedup in Non-IID Federated Bilevel Learning", ICML'23
>
> [3] "A Single-Timescale Method for Stochastic Bilevel Optimization", AISTATS'22
>
> [4] "Closing the gap: Tighter analysis of alternating stochastic gradient methods for bilevel problems", NeurIPS'21.
>
> [5] "SimFBO: Towards Simple, Flexible and Communication-efficient Federated Bilevel Learning", NeurIPS'23.
>
> [6] "Towards Gradient-based Bilevel Optimization with Non-convex Followers and Beyond", NeurIPS'21.
>
> [7] "Communication-Efficient Federated Bilevel Optimization with Global and Local Lower Level Problems", NeurIPS'23.
>
> [8] "On the Convergence of Momentum-Based Algorithms for Federated Bilevel Optimization Problems", arxiv.
>
> [9] "Direction-oriented Multi-objective Learning: Simple and Provable Stochastic Algorithms", NeurIPS'23.
>
> [10] "Communication-Efficient Federated Hypergradient Computation via Aggregated Iterative Differentiation", ICML'23.
>
>
> Q2: There are some fatal errors. In particular, this paper denotes $\overline{H}(x^t,y^{t+1}) := \mathbb{E}[\frac1m\sum_{i \in \mathcal{M}}\overline{H}_i(x^{t-\tau_i^t}, y^{t-\tau_i^t+1})$. Then, in convergence analysis, the authors directly use $x^t$ rather than $x^{t-\tau_i^t}$, e.g., the second equation when proving lemma 1. This is totally wrong.
>
> A2: The formula does not mean that $x^t=x^{t-\tau_i^t}$. Here $x^t$ and $x^{t-\tau_i^t}$ are the global models of $x$ in outer rounds $t$ and $t-\tau_i^t$, respectively, where $\tau_i^t$ is the asynchronous delay of client $i$ in the outer round $t$. In general, we have $x^t\neq x^{t-\tau_i^t}$, since the server updates the
> global model $x^t$ in each outer round. Therefore, in the second equation of Lemma 1's proof, we should use $x^t$ rather than $x^{t-\tau_i^t}$. To clarify the meaning, we have revised the formula as $\overline{H}^t := \mathbb{E}[\frac1m\sum_{i\in\mathcal{M}}\overline{H}_i(x^{t-\tau_i^t}, y^{t-\rho_i^t+1})]$. We are sorry for this confusion.

---

> > ### Author Response · Authors · 2023-11-21
> >
> > Q3: For an asynchronous algorithm, how the communication latency affects the convergence rate should be discussed.
> >
> > A3: Based on your comment, in the revised paper, we have added a discussion on the impacts of communication latency on the convergence rate (marked in blue on Page 7), discussed as follows. It can be found that the upper bound of the convergence rate is increasing as the average of the outer loop delay for all rounds and all clients increases. Although there are 4 non-vanishing terms, only the last term involves the asynchronous delay (i.e.,  $\frac1{\bar{\eta}T}\sum_{t=0}^{T-1} (\frac{3L_f\eta_t^2}{m^2}\sum_{i=1}^m\tau_i^t+\frac{12L_f\eta_t^2}{m^2}) \hat{D_f}$). When it turns to synchronous FBO (i.e., $\tau_i^t=1$), the convergence bound becomes $\frac1{\bar{\eta}T}\sum_{t=0}^{T-1}(\frac{15L_f\eta_t^2}{m})\hat{D}_f $, which matches previous works. When the asynchronous delay ( $\tau_i^t$) increases, we need to choose a smaller step size to keep a similar convergence bound. Intuitively, we need to use a smaller step size to make the delayed gradients affect less global gradients.
> >
> > In addition, in the revised paper, we have added some experiment results to evaluate the impacts of the asynchronous delay on the convergence rate (marked in blue in Section B.6 of the supplementary material). The experiment results show that the test accuracy is slightly smaller as the asynchronous delay increases, which matches our theoretical results. In addition, there is little convergence degradation due to the asynchronous delay, which proves the effectiveness and robustness of AFBO. Moreover, the experiment results show that the convergence degradation under the non-IID MINIST dataset is slightly higher than under the IID MINIST dataset. This is because the contribution of all clients is equal in the IID dataset which means the straggler has low effects on the test accuracy, but some clients owning unique datasets under non-IID dataset settings with higher asynchronous delay can cause a larger convergence degradation due to straggler effects.

---

> ### Author Response · Authors · 2023-11-22
>
> Dear reviewer: We have added responses to your comments. Could you take a look and let us know if you have any further comment? Thanks.

---

> ### Author Response · Authors · 2023-11-23
>
> Dear reviewer: We have added responses to your comments. Could you take a look and let us know if you have any further comment? Thanks.

---

> ### Comment · Area_Chair_rNtp · 2023-12-04
>
> Dear Reviewer PRRM,
>
> The authors have answered your comments. Please read their responses as soon as possible (if you haven't) and reply here that if you still think there is a fatal error. If the authors have addressed your concerns (maybe some of them), please update your rating if you have change your evaluation. Thank you so much.
>
>
> Thanks,
> Area Chair

---

### Meta-Review · Area_Chair_rNtp · 2023-12-06

**Metareview:**

This paper considers a very flexible framework of federated learning where the clients in the distributed system can participate in any inner or outer rounds, asynchronously, and with any number of local iterations.

However, this paper makes a critical error. At the beginning of Section 3.2, the authors said

$$
\nabla \Phi(x) =(1/m)\sum_{i\in\mathcal{M}} \nabla f_i(x,y^*(x)) = (1/m)\left\\{\sum_{i\in\mathcal{M}} \nabla_x f_i(x,y^*(x))  -   \nabla_{xy}^2 g_i(x,y^*(x)) \times [ \nabla_{yy}^2 g_i(x,y^*(x)) ]^{-1}  \nabla_y f_i(x,y^*(x)) \right\\}.
$$

The second equality above is wrong. Note that
$$\nabla f_i(x,y^*(x)) = \nabla_x f_i(x,y^*(x))  -   \frac{\partial y^*(x)}{\partial x}  \nabla_y f_i(x,y^*(x)) .$$

Then by implicit function theorem and the finite-sum structure of $g$, we have
$$
\frac{\partial y^*(x)}{\partial x} = \nabla_{xy}^2 g(x,y^*(x)) \times [ \nabla_{yy}^2 g(x,y^*(x)) ]^{-1}
= (1/m)\sum_{j\in\mathcal{M}} \nabla_{xy}^2 g_j(x,y^*(x)) \times\left [(1/m)\sum_{k\in\mathcal{M}} \nabla_{xy}^2 g_k(x,y^*(x))  \nabla_{yy}^2 g_k(x,y^*(x)) \right]^{-1}.
$$
Applying this to the definition of $\nabla \Phi(x)$, there exist (three-layer) nested summation over index $i,j,k$, which gives
$$
\nabla \Phi(x) =(1/m)\sum_{i\in\mathcal{M}} \nabla f_i(x,y^*(x)) = (1/m)\left\\{\sum_{i\in\mathcal{M}} \nabla_x f_i(x,y^*(x))  -   (1/m)\sum_{j\in\mathcal{M}} \nabla_{xy}^2 g_j(x,y^*(x)) \times\left [(1/m)\sum_{k\in\mathcal{M}} \nabla_{xy}^2 g_k(x,y^*(x))  \nabla_{yy}^2 g_k(x,y^*(x)) \right]^{-1} \nabla_y f_i(x,y^*(x)) \right\\}.
$$

Compared to what the authors wrote at the beginning of Section 3.2, we can see that the authors simply interchange summation with matrix inversion and ignore the double summation. I originally thought this is just a typo, but when I check the algorithm, I realize the authors really think this is true and develop the algorithm based on that. This can be reflected from Line 15 of Algorithm1, where the authors really just sum up $H_i^{t-\tau_i^t}$ which is an approximation of $\sum_{i\in\mathcal{M}} \nabla_x f_i(x,y^*(x))  -   \nabla_{xy}^2 g_i(x,y^*(x)) \times [ \nabla_{yy}^2 g_i(x,y^*(x)) ]^{-1}  \nabla_y f_i(x,y^*(x)) $.

With this big error, I have to recommend rejection.

AC

**Justification For Why Not Higher Score:**

There is a critical error.

**Justification For Why Not Lower Score:**

NA

---

### Decision · Program_Chairs · 2024-01-16

Reject